# Exploring Plant-Based Ethnomedicine and Quantitative Ethnopharmacology: Medicinal Plants Utilized by the Population of Jasrota Hill in Western Himalaya

**Bishander Singh** [1] , **Bikarma Singh** [2,3,*] , **Anand Kishor** [1] , **Sumit Singh** [2,3] ,
**Mudasir Nazir Bhat** [2,3] , **Opender Surmal** [2,3] and **Carmelo Maria Musarella** [4]

[1]  Department of Botany, Veer Kunwar Singh University, Bihar 802301, India; bishander85@gmail.com (B.S.);
    kishoranand04@gmail.com (A.K.)
[2]  Plant Sciences (Biodiversity and Applied Botany Division) CSIR-Indian Institute of Integrative Medicine,
    Canal Road, Jammu and Kashmir 180001, India; ssumitthakur14@gmail.com (S.S.);
    mudii644@gmail.com (M.N.B.); rajputop225@gmail.com (O.S.)
[3]  Academy of Scientific and Innovative Research (AcSIR), Ghaziabad 201002, India
[4]  Department of Agraria, Mediterranea University of Reggio Calabria, 89122 Reggio Calabria, Italy;
    carmelo.musarella@unirc.it
*   Correspondence: drbikarma@iiim.res.in

**Abstract:** Plants and natural products have played a significant role in curing and preventing a variety of ailments occurring in humans and animals, and continue to provide new bioactive leads for researchers in therapeutic discovery. This study was conducted with the aim to identify and document local healers' practices of treating human diseases and quantitatively document indigenous knowledge of medicinal plants, as well as to highlight the species of public interest for bioprospecting potential. A total of 17 field tours were carried out in 12 regions of Jasrota hill and its adjoining areas of Himalaya. Informants (113) were interviewed using semi-structured interviews and discussions and local guided collections. The results were analyzed using ethnobotanical indices—use-reports (URs) and the informant consensus factor (ICF)—and the data were statistically analyzed. The ethnopharmacological uses of 121 plant species belonging to 105 genera and 53 families were reported for use as medicine for treating 93 types of ailments. A total of 4987 URs were mentioned by 113 informants. Fabaceae (90.09%) and Asteraceae (6.62%) were the most represented families. Herbs (46.28%) were the primary sources of medicine, decoction (33.88%) was the most common use method for utilization, and leaves (43.80%) were the most frequently used plant parts. The ICF values ranged from 0.667 to 0.974, with the highest number of species (1314UR, 55 species) being used for the treatment of gastrointestinal ailments (GIA), followed by dermatological disorders (38 species). This result showed that the exchange of knowledge could be evident among the different communities, and their medicinal uses and practices could be correlated.

**Keywords:** ethnobotany; quantitative ethnopharmacology; Jasrota hill; Himalaya; conservation; drug discovery

---

## 1. Introduction

Plants are an integral part of life, as they produce secondary metabolites to fight disease [1,2], and humans have rationalized medicinal plants and their chemical components in various forms for therapeutic use [3,4]. Usually, conventional medicines are effective in curing many common

diseases, but complex diseases (COVID-19, AIDS, cancer, etc.) are becoming more challenging [5–7]. Ethnobotanical plants have a strong base in Indian systems of medicine, such as Ayurveda, Unani, and Siddha, and in several other local medicines such as the Chinese and Tibetan systems [8–10]. The practice of traditional medicine is not the same all over the world and varies depending on culture and philosophy [11–13]. It is estimated that traditional medicine uses more than 5000 plant species comprising different taxonomic groups [14]. Nowadays, indigenous knowledge about the use of medicinal plants in being lost from one generation to another generation due to modernization and the habitat destruction of the medicinal plants [15,16]. Therefore, the documentation of traditional knowledge plays an important role in its conservation and facilitates future medicinal plant research for the discovery of medicines [17,18].

India, internationally officially called "The Republic of India", lies within the latitudes and longitudes of 8.4–37.6° N and 68.7–97.3° E, respectively; is positioned on the Indian subcontinent in South-Central Asia; and is located in both the eastern and northern hemispheres of the globe. This biogeographic zone is ranked 7th largest country by an area of 3,287,263 km$^2$; is the second-most populous country, represented by 1,352,642,280 people (available at https://population.un.org/wpp); and ranks 6th among the total 12 mega biodiversity centers of the world.

In India, the Himalaya regions (the eastern, western, and central zones) are recognized as one of the four biodiversity hotspots (the Himalayas, Western Ghats, Indo-Burma, Sundaland), and of the reported 18,532 species of angiosperms (available at http://bsienvis.nic.in), an estimated 7000 plant species have medicinal utilization in traditional folklore and Indian systems of medicine, such as Ayurveda, Siddha, Unani, and Homeopathy [19]. This country is home to different ethnic cultures and tribes, where 22 scheduled languages (15 Indo-European, 4 Dravidian, 2 Sino-Tibetan, 1 Austroasiatic) are recognized by the Indian constitution (available at https://www.britannica.com), and different customs and systems of medicine are followed. The wild botanical plants, as natural resources used by tribal communities in several regions of India, have been the subject of ethnobotanical surveys [20], and this is due to the rich ethnic and cultural diversity that lies within this country. Research on the traditional uses of plants is mostly based on citations mentioned in ancient works of literature, such as the Ayurveda and Charaka Samhita [21]. These renowned texts highlight the value of ethnobiology in herbal drug formation. It is reported that Indian traditional systems of medicine usually employ a large number of species of plants, and the majority of them are employed in the Ayurvedic system (>2000 species); others include Siddha (1121), Unani (751), and Tibetan (337) [22].

The Himalayas are considered one of the largest mountain ecosystems of the world and are known for their rich ethnobotanical wealth, and particularly for medicinal plants since ancient times [23], because they harbor >50% of the total country's biodiversity, and most of the plants are endemic and unique [24,25]. The newly formed union territory of India, Jammu and Kashmir (J&K) (latitude of 32°17′–37°03′ N, the longitude of 72°03′–80°20′ E), has a total area of 222,236 km$^2$ and a population of 12,541,302 (available at https://censusindia.gov.in), and is a part of the Himalaya hotspot. It shares a border with Himachal Pradesh and Punjab (south), China (north and east), Pakistan (west and northwest), and is divided into three divisions (Jammu, Kashmir, Ladakh). This union territory possesses a great variation in altitude and unique climates, which vary from tropical to subtropical, temperate, alpine, and cold alpine [26], which has resulted in the evolution of a diverse group of flora and fauna [27]. The local people of J&K depend on plants for treating common diseases, which is due to the absence of modern medicinal facilities. Therefore, many researchers desire to work on ethnobiology to get access to the indigenous knowledge and culture possessed by the locals [28,29]. Other ethnomedicinal studies [30–37] were from the adjoining biogeographic regions. It has now become more important than ever to investigate and preserve traditional knowledge, and this also will aid the discovery of new drugs and herbal formulations [38,39]. Realizing that ethnomedicine knowledge is being lost, the present study planned to collect and document information on the indigenous use of medicinal plants for the treatment of common diseases by the local people

of J&K for future generations and to provide baseline information for pharmacological studies on the discovery of new drugs and molecules.

In this research, an ethnopharmacological study was conducted in the mountains and valleys of Jasrota hill and the surrounding mountainous areas, which are predominantly part of the western Himalayas and still a less studied region of the protected area within India. The documentation of traditional knowledge associated with plants was of particular interest to us, because, despite being situated in the Himalayan hotspot ecosystem, in several regions such as Jasrota hill and other interior regions comprehensive studies regarding the utilization of medicinal and otherwise important plants by the local communities—such as the *Duggars, Paharis, Punjabis*, and *Gujjars*—are still lacking. However, there are a few reports of similar studies [40–42] in the district and state from other parts of the area.

The present study focused on field study and the collection of the maximum quantitative ethnopharmacological data through discussion with local people in the hills and valleys of Jasrota and the surrounding mountains and the analysis of the data obtained. We also evaluated the most valued plant species of medicinal importance using published information. The rationale for utilizing the mentioned plants by the local population of the study area is also discussed.

## 2. Material and Methods

### 2.1. Study Area and the Local Populace

Lying between 32°27′ and 32°31′ N latitude and 75°22′ and 75°26′ E longitude, Jasrota hill or Jasrota Wildlife Sanctuary (JWS) is located on the banks of the river Ujh in the Kathua district in the Jammu and Kashmir Union Territory (UT) within the Himalayas of the Indo-Malayan biogeographical realm (Figure 1). This sanctuary and the adjoining area are recognized as compact masses of natural forest ecosystems, with a total area of 10.04 km$^2$ for the core JWS where the core belt area is virgin and undisturbed. The elevation of the JWS ranges from 356 to 650 m above the mean sea level (AMSL). Jasrota village lies to the north of this sanctuary, and the adjoining village areas have particular types of Shivalik vegetation composition. The Shivalik is the outermost range of the foothills of the Indian Himalayan belt extending to a lower elevation dominated by the Himalayan subtropical forests. Since the sanctuary is located at the junction of two biotic provinces, it shows the agro-ecological link of Indian geographical regions such as Trans-Himalaya and Himalaya. These hilly regions of Kathua have very unique and interesting biodiversity, harboring several threatened, endemic, and otherwise economically valued plants, and also represent a center of origin for a significant number of cultivated crop plants and their wild relatives. Kathua harbors a very rich flora comprising 1567 species of vascular plants, which represent 35.3% of the flora of J&K and 8.6%of that of India, including species of medicinal and aromatic plants.

About 85% of the JWS is covered by natural forests, and the major vegetation type in this sanctuary is the typical Himalayan subtropical Shivalik forest. The subtropical here includes semi-evergreen and deciduous forests. The climate in Kathua district and Jasrota wildlife sanctuary in particular varies according to altitude, and they possess typical Shivalik-type landforms. The subtropical climate with distinct seasons (summer, rainy, and winter) prevails over Jasrota and its surrounding belts of Kathua; however, the high-altitude regions of the district experience snow and occasional frost during the winter season. The mean annual temperature varies from 8 to 42 °C. The months of March to June are the summer season, and the average temperature ranges between 36 and 42 °C; July to October is the rainy season (the temperature varies from 25 to 35 °C); and November to February is categorized as the winter season, whose temperature varies from 8 to 25 °C. Rainfall in Kathua and particularly in Jasrota is a little less plentiful, varying from 200 to 300 mm in the plain, while in some parts of high altitude it varies from 500 to 1000 mm. The remaining regions have the same climate throughout the year. The extensive and densely populated region of Kathua is the home of many ethnic groups, such as the *Duggars, Paharis, Punjabis, Gujjars*, and other Indian people belonging to different races and castes. Most of the local tribes in this area are distributed in all parts, cover large

parts of J&K, and are spread across the political boundaries of India. These groups of inhabitants have a unique tradition and follow peculiar rituals. The main occupation is the collection of forest resources, and they depend on traditional farming and rain-fed cropping. The *Duggars* and *Pahari* people belong to the Indo-Aryan ethnolinguistic group living in the Himalayas, whereas the Gujjars are an ethnic pastoral group that used to be nomadic. There is no village inside the Jasrota Wildlife Sanctuary; however, there are several small villages around and nearby the JWS and on the bank of the river Ujh.

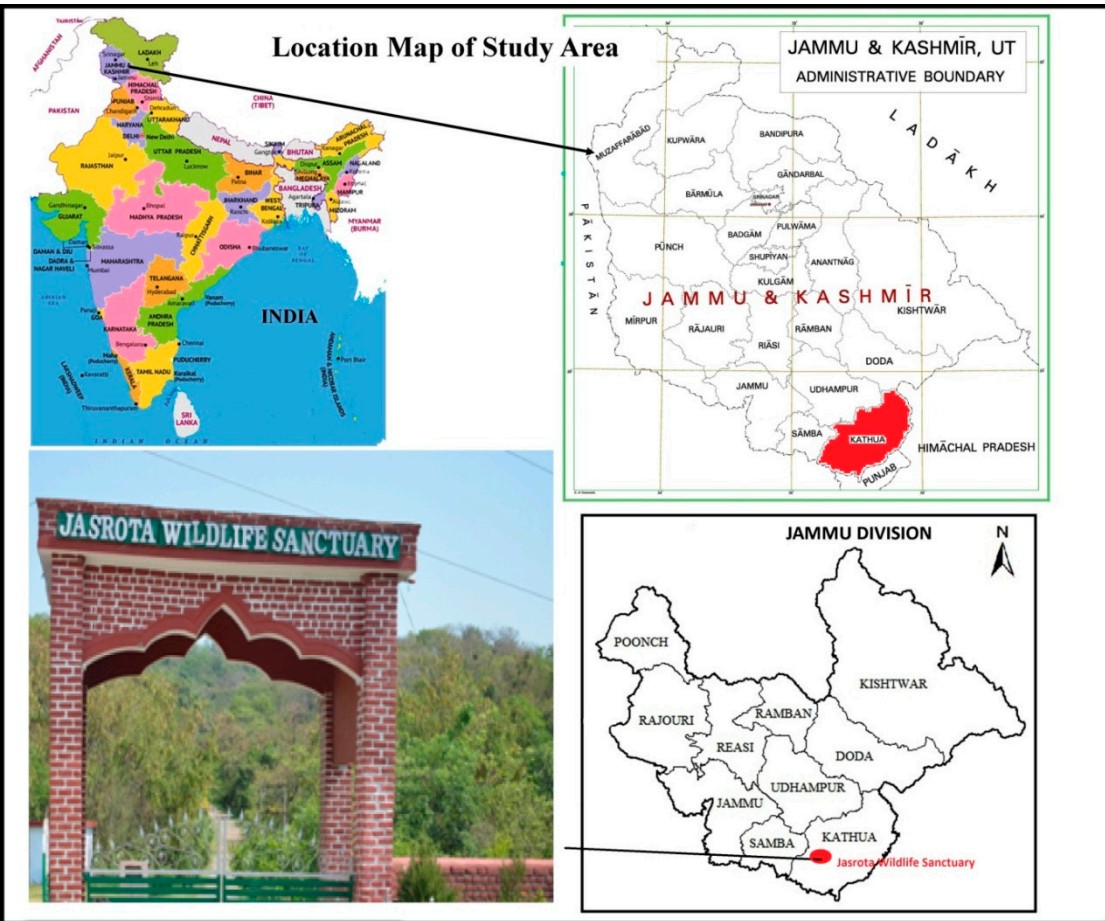

**Figure 1.** Location map of Jasrota Wildlife Sanctuary (Jasrota hill) in Jammu and Kashmir, Western Himalaya, India.

*2.2. Data Collection*

2.2.1. Field Surveys

Field trips for traditional knowledge investigation and the collection of plant samples were conducted during 2017 and 2020 in and around the protected area of 12 villages (Jasrota, Forlain, Kharot, Dhamaal, Amala, Nanan, Budhi, Merth, Jasrota Fort (inside the JWS), Shehswan, Bhorthain, Chelak), Jasrota Wildlife Sanctuary, and adjoining mountainous hills and valleys of the district Kathua. Plants of medicinal value were collected together with the informants in triplicate and the herbarium was prepared as per standard protocols [43]; the specimens of the collected plants of medicinal value were pressed, dried, and pasted on herbarium sheets (42 × 28 ± 2 cm). The specimens were deposited in the Janaki Ammal Herbarium (acronym RRLH) of CSIR-Indian Institute of Integrative Medicine Jammu. The acronym for the herbarium is in accordance with Thiers [44].

### 2.2.2. Ethnobotanical Data Collection

Sampling Design and Methods

Since the ethnopharmacological field studies mostly focused on indigenous knowledge [45], we aimed to study the people who practice self-medication, ayurvedic doctors, herbalist headmen of the targeted localities (12 villages), and aged villagers who had average knowledge of local plants. We followed the random sampling method proposed by Weckerle et al. [46] and Bolfarine and Bussab [47] to collect sufficient unbiased samples and used the probabilistic stratified randomized sampling method mentioned by Scheaffer et al. [48]. We also followed Espinosa et al. [49] for the calculation of the estimated total sample size to gain appropriate data from the whole investigated area.

Informants and Interviews with Local People

The investigation for ethnopharmacological surveys was conducted 17 times between August 2017 and March 2020 with local people belonging to the *Duggars, Paharis, Punjabis*, and *Gujjars* communities native to and presently living in the study area. A total of 113 individuals were interviewed in 12 villages (Jasrota, Forlain, Kharot, Dhamaal, Amala, Nanan, Budhi, Merth, Jasrota Fort (inside the JWS), Shehswan, Bhorthain, Chelak) (Table 1). The total population reported for these villages was 31,074, and Forlain village was the most populated village with 6462 people, followed by Chelak (4541) and Budhi (3805). These hilly villages were selected because of the plant richness culture similarly to the people, and efforts were made to include the native people who knew about the local medicinal plants and wildlife. The interviews were carried out in the local Dogri language to gather maximum knowledge and to carry out proper discussion regarding the plants. A total of nine resource people were identified to gain direct information from local traditional practitioners. The standard questionnaires were prepared and used for data collection, and the standard protocols of Martin [50], Cotton [51], and the International Society of Ethnobiology (available at https://www.ethnobiology.net) were followed.

**Table 1.** Targeted localities, the distribution of population, and the number of use-reports on the medicinal plant species of Jasrota hill and the adjoining area, Jammu and Kashmir, Western Himalaya, India.

| Locality | Distance from Jammu (km) | Total Population * | Sample (Population) Fraction | Sample Size of Proposed Informants (n) | Sample Size of Final Informants (fn) | Use-Reports Mentioned (UR) ** | Plant Species Identified | Relative Frequency (Rf,%) |
|---|---|---|---|---|---|---|---|---|
| Jasrota | 82 | 2665 | 0.0858 | 25 | 22 | 943 | 109 | 90.08 |
| Forlain | 75.3 | 6462 | 0.2080 | 11 | 11 | 489 | 59 | 48.76 |
| Kharot | 88.5 | 3549 | 0.1142 | 5 | 5 | 442 | 87 | 71.90 |
| Dhamaal | 42 | 859 | 0.0276 | 9 | 9 | 203 | 43 | 35.54 |
| Amala | 74 | 2021 | 0.0650 | 5 | 5 | 579 | 78 | 64.46 |
| Nanan | 77.6 | 768 | 0.0247 | 13 | 12 | 237 | 23 | 19.01 |
| Budhi | 76 | 3805 | 0.1224 | 10 | 9 | 212 | 29 | 23.97 |
| Merth | 73 | 1795 | 0.0578 | 8 | 8 | 398 | 67 | 55.37 |
| Jasrota fort (inside JWS) | 89 | 6 | 0.0002 | 3 | 2 | 126 | 57 | 47.11 |
| Shehswan | 79.5 | 2095 | 0.0674 | 7 | 6 | 213 | 29 | 23.97 |
| Bhorthain | 76.5 | 2508 | 0.0807 | 6 | 6 | 278 | 43 | 35.54 |
| Chelak | 44 | 4541 | 0.1461 | 18 | 18 | 867 | 101 | 83.47 |
| Total | | 31,074 | | 120 | 113 | 4987 | 121 | |

* Source: 2011 Census India (Kathua district, J&K); ** total use-reports or citations mentioned by male and female informants are 2378 and 2609.

The information was collected by conducting interviews and group discussions on the indigenous use of plant species as medicine. Selecting the people after a preliminary study, knowledge about their interests and skills in identification and utilization was obtained through informal interviews. Before group discussion, the objectives were elaborated to the informants. The semi-structured interviews were conducted with participants in the field studies, and discussion-cum-interviews were conducted face to face with the selected informants identified by the resource people targeting community aged persons, herbal healers, and youths called for participation (Table 2). The consent forms designed for the purpose were signed by the participants to show their willingness to participate. In addition, photography and voice recordings were conducted for later verification. The informants were asked to provide data on the local name of the plants, their habitat, whether the plants were wild/cultivated, the parts of the plant used, the diseases that were curable, the method of crude drug preparation and techniques, the disease and administration recipes, the mode of preparation, the side-effects (if any), and the duration of applications; these are included as a result of this study. As shown in Table 3, the emic perceptions of ailments as reported by the informants (UR, use-report) were used to analyze the cultural significance.

**Table 2.** Cartography details of the informants that participated in the study and the Spearman rank correlation analysis.

| Informants | Male | Female | Total |
|---|---|---|---|
| | 72 (67.72%) | 41 (36.28%) | 113 (100%) |
| **Age class** | **Male** | **Female** | |
| 21–30 | 11 (15.28%) | 3 (7.32%) | 14 (12.39%) |
| 31–40 | 17 (23.61%) | 9 (21.95%) | 26 (23.01%) |
| 41–50 | 22 (30.56%) | 6 (14.63%) | 28 (24.78%) |
| 51–60 | 8 (11.11%) | 13 (31.71%) | 21 (18.58%) |
| 61–70 | 5 (6.94%) | 3 (7.32%) | 8 (7.08%) |
| 71–80 | 9 (12.50%) | 7 (17.07%) | 16 (14.16%) |
| Total | 72 (100%) | 41 (100%) | 113 (100%) |
| **Literacy level** | **Male** | **Female** | |
| Never attended a school | 25 (34.72%) | 13 (31.71%) | 38 (33.63%) |
| Attended school for 1–5 classes | 17 (23.61%) | 18 (43.90%) | 35 (30.97%) |
| Attended school for 6–10 classes | 19 (26.39%) | 9 (21.95%) | 28 (24.78%) |
| Intermediate (12th class) | 7 (9.72%) | 1 (2.44%) | 8 (7.08%) |
| Graduate | 3 (4.17%) | 0 (-) | 3 (2.65%) |
| Post-graduate | 1 (1.39%) | 0 (-) | 1 (0.88%) |
| Total | 72 (100%) | 41 (100%) | 113 (100%) |
| **Spearman rank correlation** | **r** | **p** | |
| Total individuals or informants (n = 113; $\alpha$ = 0.01) | | | |
| Individuals age vs number of citations | 0.257 | <0.001 ** | |
| Male individuals (n = 72; $\alpha$ = 0.01) | | | |
| Individuals age vs citation number | 0.314 | <0.001 ** | |
| Female individuals (n = 41; $\alpha$ = 0.01) | | | |
| Individuals age vs citation number | 0.200 | <0.012 * | |

Correlation is significant at the 0.05 (*) and 0.001 (**) probability levels; n—number; *p*—probability; r—Spearman's correlation coefficient.

**Table 3.** Quantitative ethnomedicinal analysis of the 15 categories of indigenous use for commonly used plants of Jasrota hill and the surrounding mountains, Jammu and Kashmir, Western Himalaya, India.

| Ailments Categories | Important Ailments (or Disease) of a Category | $N_{ur}$ | %$N_{ur}$ | ICF | $N_t$ | %$N_t$ | %WP | %UGP | %BR | %AEP | %LR |
|---|---|---|---|---|---|---|---|---|---|---|---|
| Gastro-intestinal ailments (GIA) | constipation, abdominal spasms, acidity, dysentery, diarrhea, flatulence, vomiting, loss of consciousness, intestinal or oral ulcers, indigestion, stomachache, piles, laxative, nausea, aperients, appetizer, anthelmintic infections, halitosis | 1314 | 26.35 | 0.959 | 55 | 45.45 | 24.55 | 33.08 | 47.18 | 24.39 | 1.37 |
| Dermatological infections (DID) | cuts, itching, pimple, skin irritation, wound, abscess, acne, boils, burns, leukoderma, ringworm, leprosy, scabies, dermatitis, warts, stings of allergic plants | 946 | 18.97 | 0.961 | 38 | 31.40 | 18.67 | 24.20 | 22.58 | 17.58 | 24.66 |
| Respiratory systems diseases (RSD) | asthma, bronchitis, pneumonia, chest infection, cough, cold, influenza, pharyngitis, whooping cough | 592 | 11.87 | 0.961 | 24 | 19.83 | 12.28 | 11.49 | 4.84 | 12.50 | 6.85 |
| Skeleto-muscular system disorders (SMSD) | body pain, muscle pain, rheumatic pain, joint pain, bone fractures or bone broken, swellings in bone, arthralgia, | 482 | 9.67 | 0.958 | 21 | 17.36 | 9.08 | 4.90 | 8.87 | 10.80 | 10.96 |
| Gento-urinary ailments (GUA) | abortion, breast pain, delivery pain, lactation, over bleeding, sexual power, urine blockage or painful urination, venereal disease, male fertility, bloody urine, urinary bladder disorders, urinary stones, gynecological disorders (abortion, menstrual problems) | 352 | 7.06 | 0.949 | 19 | 15.70 | 3.96 | 1.23 | 4.84 | 8.85 | 20.55 |
| Fever (FVR) | body temperature, malaria, hay-fever, febrifuse, antipyretic, typhoid | 175 | 3.51 | 0.937 | 12 | 9.92 | 4.86 | 4.44 | 1.21 | 3.16 | 4.11 |
| Endocrinal system disorders (ESD) | diabetes, adrenal diseases, parathyroid | 137 | 2.75 | 0.971 | 5 | 4.13 | 7.16 | 0.00 | 0.40 | 2.45 | 1.37 |
| Hair care (HC) | removal of dandruffs, graying hair, hair growth, external parasites (lice, ticks), hair loss, hair tonic | 132 | 2.65 | 0.954 | 7 | 5.79 | 5.37 | 0.15 | 1.21 | 2.63 | 1.37 |
| Liver problems (LP) | jaundice, hepatitis, liver enlargement, liver pain | 132 | 2.65 | 0.954 | 7 | 5.79 | 0.90 | 5.51 | 0.00 | 2.72 | 1.37 |
| Dental care (DC) | foul odor, teeth strength, toothache, worms in gum and teeth | 118 | 2.37 | 0.974 | 4 | 3.31 | 2.17 | 0.31 | 0.40 | 2.97 | 2.74 |
| Ear, nose and eye problems (ENT) | color blindness, otalgia, earache, nasal infection, eye pain, blurred vision, conjunctivitis, eye inflammation, ophthalmia | 116 | 2.33 | 0.957 | 6 | 4.96 | 2.05 | 0.46 | 1.21 | 2.85 | 2.74 |

| Ailments Categories | Important Ailments (or Disease) of a Category | $N_{ur}$ | $\%N_{ur}$ | ICF | $N_t$ | $\%N_t$ | %WP | %UGP | %BR | %AEP | %LR |
|---|---|---|---|---|---|---|---|---|---|---|---|
| Poisonous bites (PB) | poison bites, snake bites, scorpion sting | 115 | 2.31 | 0.956 | 6 | 4.96 | 1.41 | 5.21 | 1.61 | 1.83 | 9.59 |
| Mouth disorders (MD) | mouth and tongue ulcer, thrush, leukoplakia, cold sores | 111 | 2.23 | 0.936 | 8 | 6.61 | 0.51 | 2.30 | 0.00 | 2.82 | 1.37 |
| Cancerous (CC) | cancer—lung, tumor | 19 | 0.38 | 0.667 | 7 | 5.79 | 0.26 | 0.31 | 4.84 | 0.06 | 1.37 |

Note: total number of use-reports is 4987; total number of plant species is 121; $N_{ur}$—number of use-reports; $\%N_{ur}$—percentage of use-reports contributed to the total amount of use-reports by the respective ailment category; ICF—informant consensus factor; $N_t$—number of medicinal plant species; $\%N_t$—percentage of the medicinal plant species reported for ailment category with respect to the total amount of reported plant species; %WP—percentage of use-reports for the respective ailment category which indicate the whole plants (small herbs); %UGP—percentage of use-reports for the respective ailment category that indicate underground parts such as rhizomes, tubers, roots, and bulbs; %BR—percentage of use-reports for the respective ailment category that indicate barks either as barks of stems, branches, and roots; %AEP—percentage of use-reports for the respective ailment category that indicate aerial parts in the form of leaves, young twigs, individual flowers, inflorescence, fruits, and seeds; %LR—percentage of use-reports for the respective ailment category that indicate latex and resin, including dried gums.

### 2.2.2.3. Ailment Categories

Based on the information gathered from the local healers (ayurvedic doctors, locally called vaid and hakkim) living around 12 villages of Jasrota hill, all the reported ailments from informants were categorized into 15 different categories. These includes cancerous (CC); dental care (DC); dermatological infections (DID); ear, nose, and eye problems (ENT); endocrine disorders (ESD); fever (FVR); gastro-intestinal ailments (GIA); general health care (GHC); gento-urinary ailments (GUA); hair care (HC); liver problems (LP); mouth disorders (MD); poisonous bites (PB); respiratory diseases (RSD); and skeleton-muscular disorders (SMSD). According to the body system treatment, several biomedical terms for the disease were placed/categorized in one ailment category (Table 3).

### *2.3. Data Analysis*

The ethnopharmacological data collected from field surveys were evaluated through the quantification of use-reports (URs) and the informants' consensus factor (ICF), as described by Trotter and Logan [52]. Besides this, we also evaluated the most valued medicinal plants of the study area using the published literature on the natural products, phytochemistry, and biological functions (pharmacological activities) of the studied species.

### 2.3.1. Informant Consensus Factor (ICF)

The ICF was used to analyze if there were agreements about the use of plants in ailment categories between the plant users in the investigated area, and was calculated by applying the following formula [53–55]:

$$ICF = \frac{N_{ur} - Nt}{N_{ur} - 1},$$

where $N_{ur}$ indicated the number of URs for a particular ailment category, $N_t$ refers to the total number of taxa used for a particular ailment category according to the informants, and the product factor ranges between 0 and 1 [55]. A high value towards 1 suggests that relatively few taxa are used by a large proportion of the informants, and a low value indicates that the interview informants disagree on the taxa to be used for curing ailments [56]. The ICF reflects the homogeneity of the information and consensus between the individuals [57,58].

The Spearman correlation test [59] was also studied to evaluate the correlation between the age of individuals and the number of uses mentioned for ethnomedicinal plants. The Mann–Whitney test [60] was applied to find out the variation in plant citations by male and female informants. All the analyses related to statistics were performed using SPSS 16.

2.3.2. Literature Reviews

Species Identity and Library Consultation

A total of 412 herbarium specimens of plants were prepared as a specimen record mentioned by the informants as medicine. The scientific literature on ethnomedicinal field studies conducted in the Jammu & Kashmir and Himalaya regions available in scientific journals, books, and monographs were consulted in the library. Plant species were botanically identified with the help of "Flora of Jammu and Plants of Neighbourhood" [58], "Flora of Trikuta Hills" [61], "Flora of Udhampur" [62], "Handbook of Medicinal Herbs" [63], "Indian Medicinal Plants-An Illustrated Dictionary" [64], "Illustration of Jammu Plants" [65], and "Plants for Wellness and Vigour" [66]. Books and research papers based on secondary information were not included in the present study. For the discussion on phytochemistry and pharmacological applications, a few selected species-specific review works were consulted within a use category. A few selected most popular herbal pharmacopeias of the world, such as the American herbal pharmacopeia (available at https://herbal-ahp.org), British pharmacopeia (available at https://www.pharmacopoeia.com/), "Encyclopedia on Indian Medicinal Plants" by the ENVIS center on medicinal plants, FRLHT (available at http://envis.frlht.org/), and other similar literature on Indian medicinal plants, were also consulted.

Angiosperm Phylogeny Group IV [67,68] was followed to classify the species, and the final list of the plants was checked with The Plant List (available at http://www.the plantlist.org), the International Plant Names Index (available at http://www.ipni.org), and Tropicos (available at https://www.tropicos.org). The botanical terms followed are according to Pereira and Agarez [69].

Herbarium work for the authenticity of the plant was carried out by comparing and consulting herbarium samples housed in Janaki Ammal Herbarium Jammu (acronym RRLH, according to Thiers 2020) [44]. Voucher specimens were deposited in the herbarium section, the Department of Botany, Veer Kunwar Singh University, Ara (Bihar), and some samples were deposited in the Janaki Ammal Herbarium, Jammu, for future reference. The voucher number of each plant identified is included in Table 4.

**Table 4.** Documented plants conclusively identified and reported to be used as ethnomedicine by the local people of Jasrota hill and the surrounding mountains, J&K, Western Himalaya, India.

| Botanical Name (Family)/Voucher Specimen Number | Vernacular Name | Life-Form | Part Used | Use-Reports (UR) | Disease Category: Ailments Treated (No. of Use-Reports Included for Particular Ailment Only) | Mode of Usage | Nativity |
|---|---|---|---|---|---|---|---|
| *Abrus precatorius* L. (Fabaceae)/10103 | rati | L | Lv | 43 | SMSD:15 (bone fracture) RSD: 27 (cough, cold) MD: 8 (tongue ulcer) | decoction | N |
| *Acacia catechu* Willd. (Fabaceae)/10106 | khair | T | W | 29 | MD:23 (mouth ulcer) GIA: 16 (dysentery) As additive: 2 (UR) | decoction | N |
| *Acacia nilotica* Willd. (Fabaceae)/10109 | kikar | T | B | 19 | GIA:7 (pile cure) DID:12 (skin diseases) | paste | N |
| *Achyranthes aspera* L. (Amaranthaceae)/10108 | puthkanda | H | R | 126 | DC:66 (relieve toothache) LP: 28 (jaundice) PB:39 (snakebites) GUA: 37 (gynecological disorders) | chew | N |
| *Acorus calamus* L. (Acoraceae)/10074 | breen | H | Rh | 99 | GIA:77 (intestinal worms, acidity) LP:10 (liver pain, liver enlargement) RSD:12 (cough, heart infection) | powder | N |
| *Adiantum capillus-veneris* L. (Adiantaceae)/10104 | hansraj | H | Rh | 103 | GUA:27 (herpes) DID:76 (wound, abscess, acne, boils, burns) | paste | N |

**Table 4.** *Cont.*

| Botanical Name (Family)/Voucher Specimen Number | Vernacular Name | Life-Form | Part Used | Use-Reports (UR) | Disease Category: Ailments Treated (No. of Use-Reports Included for Particular Ailment Only) | Mode of Usage | Nativity |
|---|---|---|---|---|---|---|---|
| *Aegle marmelos* (L) Corr. (Rutaceae)/10102 | bill | T | Lv | 35 | LP:9 (cure jaundice) RSD:3 (asthma) ESD:23 (diabetes and associated disorders) | decoction | N |
| *Aerva sanguinolenta* (L.) Blume (Amaranthaceae)/10027 | bui | S | R | 28 | ENT:28 (ophthalmic infection in goats) | juice | N |
| *Ageratum conyzoides* L. (Asteraceae)/10029 | jungli pudina | H | Lv | 69 | DID:69 (stop bleeding of cut andwound) | paste | E |
| *Albizia lebbeck* (L.) Benth. (Caesalpinaceae)/10111 | sirin | T | B | 6 | GUA:6 (impotency) | powder | E |
| *Aloe vera* (L.) Burm.f. (Xanthorrhoeaceae)/10112 | sotthu katthalai | H | Lv | 113 | DID:113 (skin injuries, wounds, dermatitis) | paste | E |
| *Alternanthera pungens* Kunth (Amaranthaceae)/10107 | khaki | H | Wp | 17 | HC:7 (hair tonic) RSD:10 (tight chest, bronchitis, asthma and other lung troubles) | decoction | E |
| *Amaranthus spinosus* L. (Amaranthaceae)/10114 | kandiari | H | Lv | 12 | GIA:12 (laxative properties) | chew | E |
| *Anagallis arvensis* L. (Primulaceae)/10117 | kokoon | H | Wp | 32 | SMSD:15 (gout) DID:17 (dermatitis) | decoction | E |
| *Argemone mexicana* L. (Asteraceae)/10121 | peelikandiari | H | Lv | 18 | DID:7 (ringworm) RSD:11 (cough) | decoction | E |
| *Artemisia scoparia* Waldst. & Kit. (Asteraceae)/10110 | jhau | H | Wp | 38 | LP:12 (jaundice, hepatitis) GIA:28 (inflammation of gall bladder) | juice | E |
| *Asparagus adscendens* Roxb. (Asparagaceae)/10086 | sanspod | L | S | 47 | GUA:9 (urinary stones, menstrual problems) GHC:27 (tonic) As additive:11 (UR) | decoction | N |
| *Azadirachta indica* A.Juss (Meliaceae)/10123 | nim | T | Lv | 71 | GIA:27 (stomach ailments, intestinal worms) LP: 11 (jaundice) DID: 19 (skin irritation) DC: 46 (tooth strength, worms in gum) | juice | N |
| *Barleria cristata* L. (Acanthaceae)/10115 | jhinti, | S | Lv | 19 | RSD:13 (cough) SMSD:6 (reducing swellings) | decoction | E |
| *Bauhinia variegata* L. (Caesalpinaceae)/10118 | kared | T | Fl | 34 | GIA:34 (dysentery, diarrhea and piles) | powder | N |
| *Boerhavia diffusa* L. (Nyctaginaceae)/10076 | lit-sit | H | R | 109 | LP:44 (jaundice) GUA:9 (nocturnal emission, urine stone) GHC:67 (memory enhancement) ESD:8 (adrenal problem) GIA:29 (constipation, stomachache) | decoction | N |
| *Bombax ceiba* L. (Malvaceae)/10120 | simbal | T | R | 19 | GIA:19 (diarrhea) | decoction | N |
| *Buddleja asiatica* Lour. (Scrophulariaceae)/10119 | neemda | S | Lv | 37 | DID:19 (skin infections) FVR:17 (malaria) CC:2 (cancer) | paste | N |
| *Butea monosperma* Taub. (Fabaceae)/10124 | dhak | T | Fl | 24 | GUA:8 (urine blockage) DID:12 (skin problem) ENT:4 (eye inflammation) | powder | N |
| *Calotropis procera* R.Br. (Ascelpiadaceae)/10127 | desi ak | S | R | 46 | DID:29 (skin diseases) SMSD:17 (joint pain, arthralgia) | paste | E |

**Table 4.** *Cont.*

| Botanical Name (Family)/Voucher Specimen Number | Vernacular Name | Life-Form | Part Used | Use-Reports (UR) | Disease Category: Ailments Treated (No. of Use-Reports Included for Particular Ailment Only) | Mode of Usage | Nativity |
|---|---|---|---|---|---|---|---|
| *Cannabis sativa* L. (Cannabaceae)/10122 | bhang | H | Lv | 47 | PB:13 (poisonous insect bites)<br>GUA:34 (pain relief) | decoction | E |
| *Capsella bursa-pastoris* Medik (Brassicaceae)/10125 | kralmond | H | Wp | 13 | GUA:5 (bloody urine)<br>GIA:8 (diarrhea) | juice | E |
| *Cardamine hirsuta* L. (Brassicaceae)/10116 | hairy bittercress | H | Lv | 19 | CC:14 (cancer)<br>GHC:12 (tonic) | juice | N |
| *Carissa carandas* L. (Apocynaceae)/10128 | kalakki | S | Wp | 38 | GIA:38 (stomachache) | decoction | N |
| *Carissa spinarum* L. (Apocynaceae)/10036 | garna | S | F | 24 | GIA:24 (stomach stones and piles) | powder | E |
| *Cassia fistula* L. (Caesalpinaceae)/10011 | karangal | T | Lv | 91 | RSD:2 (pneumonia)<br>GIA:67 (constipation, diarrhea, flatulence)<br>DID:17 (skin infections)<br>RSD:16 (cough, bronchitis) | decoction | N |
| *Cassia occidentalis* L. (Caesalpinaceae)/10126 | kasundi | S | S | 31 | GIA:17 (aperients, constipation)<br>As additive:14 (UR) | powder | E |
| *Cassia tora* L. (Caesalpinaceae)/10129 | lohki | H | Lv | 43 | FVR:13 (fever)<br>GIA:16 (stomachache)<br>DID:18 (ringworm, leprosy)<br>PB:2 (snake bite) | decoction | E |
| *Catharanthus rosea* (L.) G.Don (Apocynaceae)/10132 | sadabahar | H | Lv | 16 | DID:8 (skin disorders)<br>GHC:9 (memory enhancement)<br>CC:15 (cancer) | juice | E |
| *Centella asiatica* (L.) Urb. (Apiaceae)/10015 | brahmi | H | Lv | 57 | GHC:17 (brain tonic, blood purifier)<br>DID:48 (skin problems) | juice | N |
| *Cissampelos pareira* L. (Menispermaceae)/10040 | battal bel | L | Lv | 58 | GHC:5 (brain tonic for improving memory)<br>PB:17 (all types poisonous bites)<br>DID:16 (skin infections, skin ulcer)<br>FVR:23 (fever)<br>GUA:19 (urinary problems) | decoction | N |
| *Citrus medica* L. (Rutaceae)/10130 | gargal | T | Lv | 53 | GIA:53 (diarrhea, stomachache) | powder | N |
| *Colebrookia oppositifolia* Sm. (Lamiaceae)/10056 | duss | S | Lv | 77 | DID:77 (wound, cut and bruises) | paste | N |
| *Convolvulus arvensis* L. (Convolvulaceae)/10134 | hiran padi | H | Lv | 11 | DID:11 (skin itching) | paste | N |
| *Cuscuta reflexa* Roxb. (Convolulaceae)/10131 | andal | L | Wp | 87 | DID:32 (skin itching, wound)<br>GUA:39 (urinary problems)<br>GHC:16 (health tonic) | decoction | N |
| *Cynodon dactylon* (L.) Pers (Poaceae)/10071 | khabbal | H | R | 33 | GIA:3 (piles)<br>DID:5 (wound)<br>GHC:18 (brain tonic)<br>As additive:7 (UR) | decoction | N |
| *Cyperus rotundus* L. (Cyperaceae)/10135 | dila | H | R | 67 | PB:37 (antidote for all poisons)<br>FVR:22 (malaria)<br>GUA:8 (menstrual problems) | decoction | N |
| *Dalbergia sissoo* Roxb. (Fabaceae)/10138 | taail | T | Lv | 8 | GUA:8 (menstrual disorders) | decoction | N |
| *Daphne oleoides* Schreb (Thymelaeaceae)/10140 | kochhad | S | Lv | 19 | DID:19 (abscess) | paste | N |

**Table 4.** *Cont.*

| Botanical Name (Family)/Voucher Specimen Number | Vernacular Name | Life-Form | Part Used | Use-Reports (UR) | Disease Category: Ailments Treated (No. of Use-Reports Included for Particular Ailment Only) | Mode of Usage | Nativity |
|---|---|---|---|---|---|---|---|
| *Datura metel* L. (Solanaceae)/10137 | umatthai | H | Lv | 47 | ENT:27 (earache) GUA:20 (pain, stomachache) | juice | E |
| *Dioscorea deltoidea* Wall ex Kunth. (Dioscoreaceae)/10133 | kins | L | Lv | 42 | DID:27 (wounds, burns) SMSD:15 (swelling) | paste | N |
| *Eclipta prostrata* L. (L.) (Asteraceae)/10136 | bhangra | H | Wp | 116 | HC:21 (hair tonic) LP:47 (liver problems) RSD: 69 (asthma, cough) DID:27 (skin disorders) | powder | N |
| *Euphorbia helioscopia* L. (Euphorbiaceae)/10139 | dudhal-patal | H | La | 12 | DID:12 (abscess) | paste | E |
| *Euphorbia hirta* L. (Euphorbiaceae)/1006 | dudii | H | Wp | 115 | GIA:13 (piles) RSD:77 (cough, bronchitis, asthma) GIA:26 (dysentery, digestive disorders) | powder | N |
| *Evolvulus alsinoides* L. (Convolvulaceae)/10146 | neeli Santh | H | Wp | 23 | GHC:23 (brain tonic) | powder | N |
| *Ficus hispida* L.f. (Moraceae)/10148 | kharkhumbal | T | F | 13 | GUA:13 (galactagogue, venereal disease) | powder | N |
| *Ficus palmata* Forsk. (Moraceae)/10154 | phagara | T | F | 15 | GIA:7 (constipation) SMSD:8 (bone inflammation) | powder | N |
| *Ficus racemosa* L. (Moraceae)/10151 | rumbal | T | F | 36 | GIA:28 (kidney problems) SMSD:8 (bone inflammation) | powder | N |
| *Ficus religiosa* L. (Moraceae)/10144 | bar, peepal | T | S | 44 | GIA:29 (piles) SMSD:17 (bone inflammation) | decoction | N |
| *Galium aparine* L. (Rubiaceae)/10143 | khorti | H | Wp | 33 | GIA:33 (laxative properties) | juice | N |
| *Geranium wallichianum* Sweet. (Geraniaceae)/10152 | laal jehari | H | Lv | 29 | GIA:29 (diarrhea, dysentery) | decoction | N |
| *Hedera nepalensis* K.Koch (Araliaceae)/10135 | bano | H | Lv | 23 | SMSD:13 (arthralgia) RSD:7 (bronchitis) GUA:4 (diuretic) | juice | N |
| *Hibiscus rosa-sinensis* L. (Malvaceae)/10147 | gudaal | S | Fl | 47 | HC:27 (alopecia) GUA:14 (diuretic) | paste | N |
| *Holarrhena antidysenterica* Wall. ex A.DC. (Apocynaceae)/10142 | kagar | T | B | 44 | GIA:35 (dysentery) SMSD:14 (anti-spasmodic) | decoction | N |
| *Ipomoea carnea* Jacq. (Convolvuaceae)/10145 | bilaitti Aak | S | Lv | 51 | SMSD:51 (joint pain) | paste | E |
| *Jasminum humile* L. (Oleaceae)/10141 | sanairad | L | R | 23 | DID:13 (ring worm) ESD:10 (reduce blood sugar) | decoction | E |
| *Justicia adhatoda* L. (Acanthaceae)/10031 | brenkar | S | R | 104 | RSD:77 (cough, asthma, bronchitis) CC:1 (cancer) As additive: 26 (UR) | juice | N |
| *Lannea coromandelica* (Houtt.) Merr. (Anacardiaceae)/10150 | kamble | T | Lv | 41 | DID:24 (skin disease) MD:15 (mouth ulcer) DC:13 (toothache) | chew | N |
| *Lantana camara* L. (Verbenaceae)/1002 | panjphulli | S | La | 43 | HC:10 (promote hair growth) RSD:33 (asthma, bronchitis) DID:13 (skin itching) | decoction | E |

**Table 4.** *Cont.*

| Botanical Name (Family)/Voucher Specimen Number | Vernacular Name | Life-Form | Part Used | Use-Reports (UR) | Disease Category: Ailments Treated (No. of Use-Reports Included for Particular Ailment Only) | Mode of Usage | Nativity |
|---|---|---|---|---|---|---|---|
| *Lawsonia inermis* L. (Lythraceae)/10145 | mahendi | S | Lv | 38 | GIA:23 (stomachache) DID:17 (leukoderma, skin care) | juice | E |
| *Linum usitatissimum* L. (Linaceae)/10153 | alsi | H | S | 47 | DID:16 (abscess) GIA:29 (constipation) | paste | E |
| *Lotus corniculatus* L. (Fabaceae)/10158 | sadai | H | Wp | 39 | GHC:15 (tonic) GIA:27 (nauseaand vomiting) | juice | E |
| *Mallotus philippensis* Muell.Arg. (Eurphorbiaceae)/1005 | kamala | T | F | 36 | GIA:23 (killing worms instomach; constipation) CC:25 (cancer) | powder | N |
| *Malvastrum cormandelianum* (L.) Garcke (Malvaceae)/10161 | baddi Bareaar | H | Lv | 39 | DID:39 (styptic, skin infections) | paste | E |
| *Medicago lupulina* L.(Fabaceae)/10159 | sareri | H | Lv | 17 | GIA:17 (constipation) | powder | E |
| *Melilotus indica* All. (Fabaceae)/10063 | pili senji | H | Lv | 67 | GIA:22 (aperient) DID:47 (skin aching) | decoction | E |
| *Mentha arvensis* L. (Lamiaceae)/10155 | pootna | H | Lv | 106 | GIA:66 (flatulence, aperients, indigestion, acidity) SMSD:45 (abdominal spasms, joint pain, rheumatism) | decoction | E |
| *Mentha longifolia* (L) Huds (Lamiaceae)/10164 | jangali pootna | H | Lv | 119 | GIA:36 (flatulence, aperients) SMSD:84 (abdominalspasms, rheumatism) | decoction | N |
| *Millingtonia hortensis* L.f. (Bignoniaceae)/1007 | buchade jhad | T | Lv | 19 | FVR:7 (antipyretic) DID:3 (sinusitis) GIA:5 (cholagogue) GHC:7 (tonic) | juice | N |
| *Morus alba* L. (Moraceae)/10157 | toot | T | Lv | 38 | GIA:14 (aperients) LP:32 (jaundice) | juice | E |
| *Murraya koenigii* (L.) Spreng. (Rutaceae)/10163 | karuveppilai | T | Lv | 28 | ENT:8 (eye pain) GIA:27 (dysentery, vomiting) | decoction | N |
| *Nerium indicum* Mill. (Apocynaceae)/10166 | lal kaneer | S | Lv | 13 | RSD:13 (heart ailments) CC:1 (cancer) | juice | E |
| *Nicotiana plumbaginifolia* Viv. (Solanaceae)/10160 | desi Tamakoo | H | Wp | 48 | GIA:48 (external parasites) | paste | E |
| *Nyctanthes arbor-tristis* L. (Oleaceae)/10156 | haar-shringaar | S | Lv | 13 | GIA:13 (intestinal worms) | juice | N |
| *Oenothera rosea* Soland (Onagraceae)/10162 | darraati | H | Lv | 44 | GUA:44 (renal colic) | decoction | E |
| *Oxalis corniculata* L. (Oxalidaceae)/10187 | khattibooti | H | Lv | 48 | DC:33 (toothache) GIA:15 (halitosis) | chew | E |
| *Phoenix sylvestris* (L.) Roxb.(Areaceae)/10066 | khaajuri | T | Lv | 6 | RSD:2 (heart complaints) FVR:1 (fever) GIA:3 (vomiting, loss of consciousness) | juice | E |
| *Phyllanthus amarus* Schum & Thonn. (Euphorbiaceae)/10191 | keelanelli | H | Wp | 87 | GIA:83 (jaundice, diarrhea, dysentery) MD:7 (peptic ulcer) | decoction | N |
| *Phyllanthus emblica* L. (Euphorbiaceae)/10195 | aamla | T | B | 90 | GIA:35 (constipation) HC:13 (hair loss) GHC:48 (tonic as rejuvenator) | powder | N |
| *Physalis minima* L. (Solanaceae)/10189 | pataka | H | Lv | 23 | ENT:23 (otalgia) | juice | E |

**Table 4.** *Cont.*

| Botanical Name (Family)/Voucher Specimen Number | Vernacular Name | Life-Form | Part Used | Use-Reports (UR) | Disease Category: Ailments Treated (No. of Use-Reports Included for Particular Ailment Only) | Mode of Usage | Nativity |
|---|---|---|---|---|---|---|---|
| *Pinus roxburghii* Roxb. (Pinaceae)/10064 | chir-pine | T | Re | 19 | DID:19 (boils, cuts, wounds) | paste | N |
| *Pogostemon benghalensis* (Burm.f.) Ktze. (Lamiaceae)/10194 | kali Suaali | S | Lv | 25 | GIA:14 (dyspepsia) RSD:11 (cold, cough) | decoction | N |
| *Pogostemon plectranthoids* Desf. (Lamiaceae)/10004 | thekkali | S | Lv | 32 | GIA:19 (vomiting, diarrhea) RSD:13 (cold) SMSD:5 (headache) | decoction | N |
| *Polygonum amplexicaule* D.Don. (Polygonaceae)/10193 | masloon | H | R | 13 | RSD:13 (cold, cough) | chew | N |
| *Polygonum hydropiper* L. (Polygonaceae)/10192 | pipli | H | Wp | 28 | GIA:19 (diarrhea, piles) GUA:11 (painful menstruation, over bleeding) | decoction | N |
| *Prunus persica* (L.) (Rosaceae)/10188 | aru | T | Lv | 33 | DID:33 (cut, wound, burns, boils to soothe inflammation) | paste | E |
| *Punica granatum* L. (Punicaceae)/10190 | darunni | T | F | 47 | LP:46 (jaundice) GHC:7 (tonic) | powder | N |
| *Pyrus pashia* Buch.-Ham. ex D.Don (Rosaceae)/10051 | batangi | T | F | 26 | ENT:26 (eye infection) | juice | E |
| *Ranunculus arvensis* L. (Ranunculaceae)/10165 | charmula | H | Wp | 24 | GIA:14 (diarrhea) FVR:7 (fever) RSD:3 (asthma) | juice | N |
| *Ranunculus laetus* Wall. (Ranunculaceae)/10172 | darrili | H | La | 6 | HC:6 (hair growth) | juice | E |
| *Ranunculus muricatus* L. (Ranunculaceae)/10037 | korkhand | H | Wp | 26 | FVR:15 (periodic fever) SMSD:1 (arthralgia) RSD:13 (asthma) | juice | N |
| *Ricinus communis* L. (Euphorbiaceae)/10174 | arandi | S | Lv | 42 | SMSD:42 (headache) | paste | E |
| *Robinia pseudoacacia* L. (Fabaceae)/10176 | kikkar | T | Lv | 27 | GIA:27 (acidity, indigestion) | juice | E |
| *Rosa indica* L. (Rosaceae)/10178 | gulab | S | Fl | 19 | GIA:19 (indigestion, flatulence) | decoction | N |
| *Rubus ellipticus* Sm. (Rosaceae)/10061 | aakhey | S | Fr | 31 | GIA:15 (aperients) MD:17 (oral ulcer) | juice | N |
| *Salvia moorcroftiana* Wallich ex Benth. (Lamiaceae)/10180 | kali jadi | H | R | 19 | RSD:5 (cold) GIA:10 (stomachache and dysentery) FVR:5 (fever) | powder | N |
| *Sida cordifolia* L. (Malvaceae)/10042 | vandhamni | H | Wp | 11 | GIA:11 (stomachache) | decoction | N |
| *Sida spinosa* L. (Malvaceae)/10168 | gulsakari | H | Lv | 37 | SMSD:37 (demulcent, irritation) | decoction | E |
| *Solanum nigrum* L. (Solanaceae)/10173 | kaayankothi | H | Lv | 18 | FVR:5 (antiphlogistic) GIA:3 (stomach ulcer) DID:17 (wound) | paste | E |
| *Stellaria media* (L.) Vill. (Caryophyllaceae)/10062 | marmiri | H | Wp | 9 | SMSD:9 (swelling, bone fracture) | paste | E |
| *Synedrella nodiflora* (L.) Garetn. (Asteraceae)/10167 | jari | H | Lv | 11 | DID:11 (styptic) | decoction | E |
| *Syzygium cumini* (L.) Skeels (Myrtaceae)/10030 | jaamnoo | T | Lv | 26 | ESD: 24 (diabetes) CC:3 (cancer) | decoction | N |

**Table 4.** *Cont.*

| Botanical Name (Family)/Voucher Specimen Number | Vernacular Name | Life-Form | Part Used | Use-Reports (UR) | Disease Category: Ailments Treated (No. of Use-Reports Included for Particular Ailment Only) | Mode of Usage | Nativity |
|---|---|---|---|---|---|---|---|
| *Tamarindus indica* L. (Caesalpinaceae)/10175 | Lmli | T | F | 29 | PB:29 (anorexia, poisonous weed intake) | decoction | N |
| *Taxodium distichum* (L.) Rich (Cupressaceae)/10171 | - | T | Re | 6 | DID:6 (analgesic for wounds) | paste | E |
| *Tephrosia purpurea* (L.) Pers (Fabaceae)/10181 | sarphank | H | R | 43 | FVR:43 (typhoidfever) | powder | N |
| *Tinospora cordifolia* (Wild.) Hook.f. & Thomson (Memispermaceae)/10185 | Gloe | L | R | 105 | FVR:27 (hay fever) ESD:69 (diabetes) LP:12 (hepatitis) SMSD:33 (joint pain, rheumatism) | powder | N |
| *Toona ciliata* M.Roem (Meliaceae)/10170 | Toon | T | Lv | 9 | DID:5 (astringent) GHC:7 (tonic) | decoction | N |
| *Tribulus terrestris* L. (Zygophyllaceae)/10184 | pakhra, | H | F | 56 | GUA:37 (impotency) SMSD:21 (bone swelling) | powder | N |
| *Trifolium pratense* L. (Fabaceae)/10186 | gagar luth | H | Wp | 37 | SMDS:37 (anti-spasmodic) | decoction | N |
| *Verbascum thapsus* L. (Scrophulariceae)/10179 | soottamakoo | H | F | 38 | RSD:38 (cough, asthma, pneumonia) | juice | N |
| *Vernonia arborea* Buch.Ham (Asteraceae)/10012 | vernonia | T | B | 14 | MD:14 (mouth, tongue ulcers) | paste | N |
| *Viola odorata* L. (Violaceae)/10169 | banaksha | H | Fl | 67 | RDS:67 (cough, bronchitis, cold) | decoction | N |
| *Vitex negundo* L. (Verbenaceae)/10182 | Bana | S | Fl | 88 | GIA:27 (diarrhea, anthelminthic) RDS:57 (cough) SMSD:21 (bone pain, rheumatism) | juice | N |
| *Woodfordia fruticosa* Kurz (Lythraceae)/10177 | Dhai | S | Fl | 49 | DID:24 (skin disease) SMSD:5 (headache) GIA:29 (diarrhea, dysentery) | decoction | N |
| *Xanthium strumarium* L. (Asteraceae)/10044 | Jojra | S | Lv | 17 | SMSD:17 (headache) | decoction | E |
| *Ziziphus nummularia* (Burm.f) Wight & Arn. (Rhamnaceae)/10183 | jhar beri | S | F | 57 | GIA:57 (diarrhea, dysentery, colic) | powder | N |

Electronic Databases

The Google search (available at https://www.google.com) was carried out to search for worldwide-disseminated data using databases such as SciFinder® (available at https://www.cas.org/products/scifinder), Biological Abstract (available at https://www.ebsco.com/products/research-databases/biological-abstracts), Pubmed (available at https://www.ncbi.nlm.nih.gov/pubmed), WorldCat (available at https://www.worldcat.org), MOLBASE (available at https://www.molbase.com), Chemical Abstracts (available at https://guides.lib.utexas.edu/chemistry/chemabs), Directory of Open Access Journals-DOAJ (available at https://doaj.org/), Science Direct (available at https://www.sciencedirect.com), Web of Science Group (available at https://mjl.clarivate.com), Centre for Agriculture and Bioscience international-CABI (available at https://www.cabi.org/), and The Scientific Electronic Library Online-SciELO (available at https://scielo.org/en). Most of the searches were carried out using keywords such as the botanical name of the plant and family and the vernacular and common names. Scientific names and abbreviations of the authors of plant species mentioned here are in accordance with The Plant List, the International Plant Names Index, and the Tropicos database.

## 3. Results and Discussion

### 3.1. Characteristics of Informants

A total of 113 informants (72 males and 41 female informants) between the age group ranges of 21 to 80 years participated and were interviewed with a questionnaire. The majority (66.37%) of the informants interviewed were between the ages of 31 and 60 years. Most of them (34.72% males and 31.71% females) were poor in education and had never attended a school; however, some informants attended school for 1–5 classes (23.61% males, 43.90% females), attended school for 6–10 classes (26.39% males, 21.95% females), were intermediate (9.72% males, 2.44% females), were graduates (4.17% males, no females), and were post-graduates (1.39% males, no females) (Tables 1 and 2). Out of the total 113 informants, the lower percentage of female informants (36.28%) was because the social setup in the study area does not allow women or girls to move out of their household or participate in any outside events. The total citations of all the medicinal plants collected were 4987, and average citations per informant on the medicinal uses of plants were 44.13; the average citations of male vs. female informants were 33.03 and 63.64, respectively. The relative knowledge (citation) of informants vs. the informants' age of medicinal plants recorded from the study area in the form of linear correlation is given in Figure 2A–C. The null hypothesis ($H_o2$) is rejected at the 5% level only for informant knowledge of all uses and medicinal users, although most of the regression lines have a positive gradient. For most of the variables, however, age does not explain much of the variance in the relative informant knowledge, and these data are inadequate to conclusively test the hypothesis. Further, this study also observed that the Mann–Whitney test, comparing the traditional knowledge between the two sexes, did not show a significant decrease. Spearman's rank correlation analysis between age and the number of citations of medicinal plants ($r = 0.257$; $p < 0.001$) was significant for the investigated people. It has been observed that the older individuals, who were mostly illiterate, mentioned the maximum number of the use of plants species in comparison to the younger participants; the reason for this could be that the education of informants was negatively correlated with the number of plants reported, and that, as the education of the informants got better, they started moving away from the traditional medicinal system. In Figure 3A–C, the study depicts the relative knowledge of informants (citations of medicinal plants) vs. the level of education gathered on medicinal plants in the study area. The $R^2$ value (0.545) concerning the relative knowledge and level of education is relatively less in the case of females as compared to males ($R^2 = 0.885$) and all users ($R^2 = 0.975$).

In this study, the local inhabitants possess good knowledge of the plants, and the utilization of herbal medicine in daily life has great importance, but for the last few decades indigenous knowledge has started declining day by day among the new generation of people. Studies have reported that the ethnomedicinal plant knowledge, use, and traditional practices increase with age, and tribal people with an age above 50 years are the main custodians [70–72]. The illiterate informants are more knowledgeable than those who are literate, and a significant negative correlation was found between the number of medicinal plant reports and the education level in the study area. Similar results were reported by other ethnobiologists from surrounding areas and adjoining countries [73–77]. In the last few decades, the majority of the people have become literate (educated) and have been more exposed to modernization, and this has caused the threatening of traditional knowledge in children and younger people, as they had little interest in learning [78,79]. In the present study, it has been observed that indigenous knowledge has not been transferred from the old generation to the young generation in every household due to the lack of interest in practicing it and the ease of getting modern facilities. This is not a good sign for the young generation because, in time to come, their unique traditional knowledge may become extinct, and this is also mentioned in similar studies conducted in other parts [80–82]. Further, Rao et al. [83] believed that the education of the informants in Kathua district and adjoining areas negatively correlated with knowledge; education increases the knowledge of modern sciences, and slowly and steadily decreases the indigenous practice of culture and the use of herbal plants. Bhatia et al. [84] and Singh et al. [85] suggested that the reason for the continuously

declining indigenous medicinal plant use knowledge could be that, in order to acquire education, the younger generation moves away from their residence; after getting an education, they look for employment in cities/towns, and as time passes, they get more exposed to modernization, which could include more education, high-quality occupation, and the habit of consumerism. Similar results were also supported by other research across the globe [86].

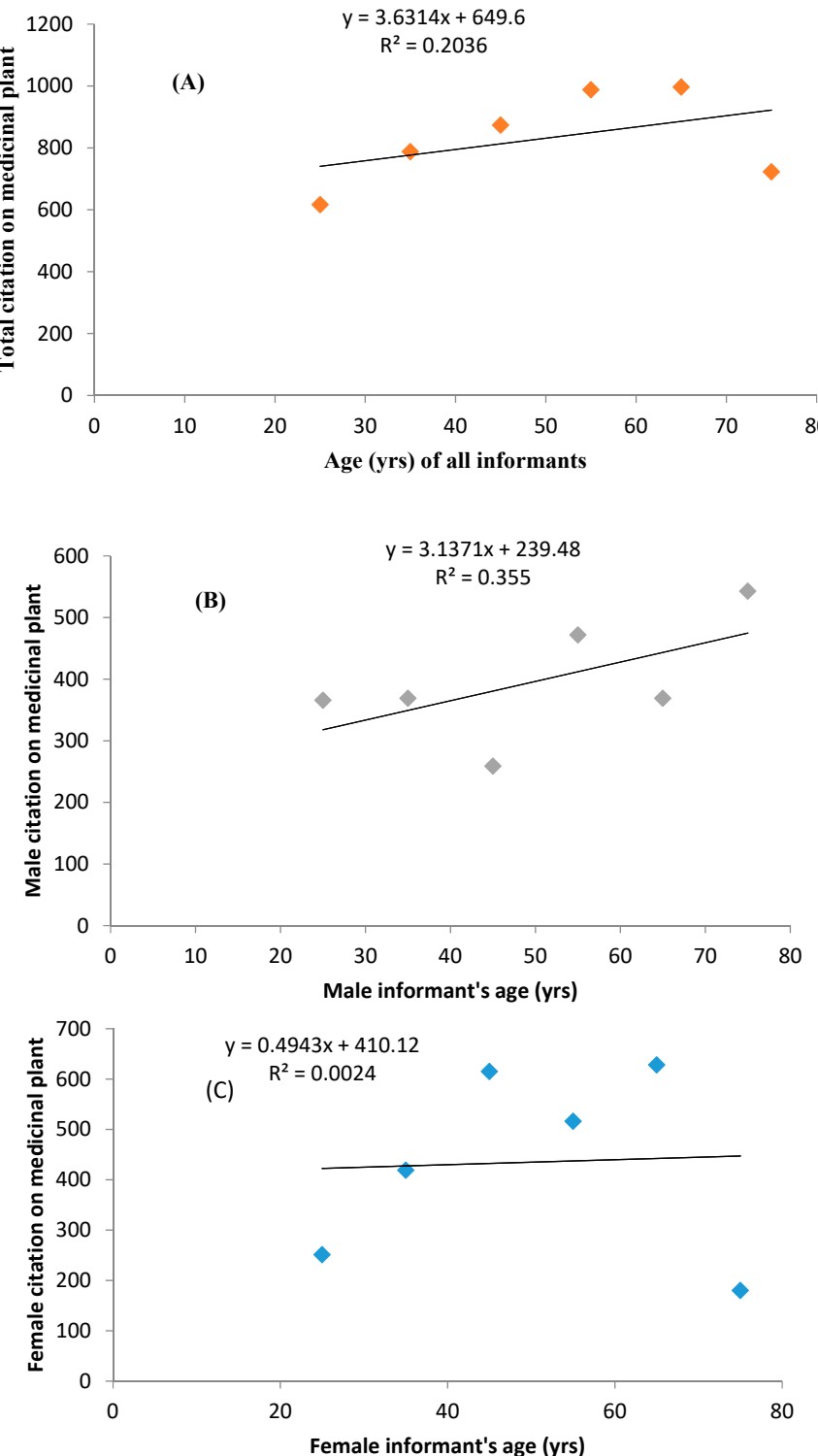

**Figure 2.** (**A**–**C**) Linear correlation—relative knowledge (citation) of informants vs. informants' age of medicinal plants in the study area: (**A**) all users, (**B**) male informants, (**C**) female informants.

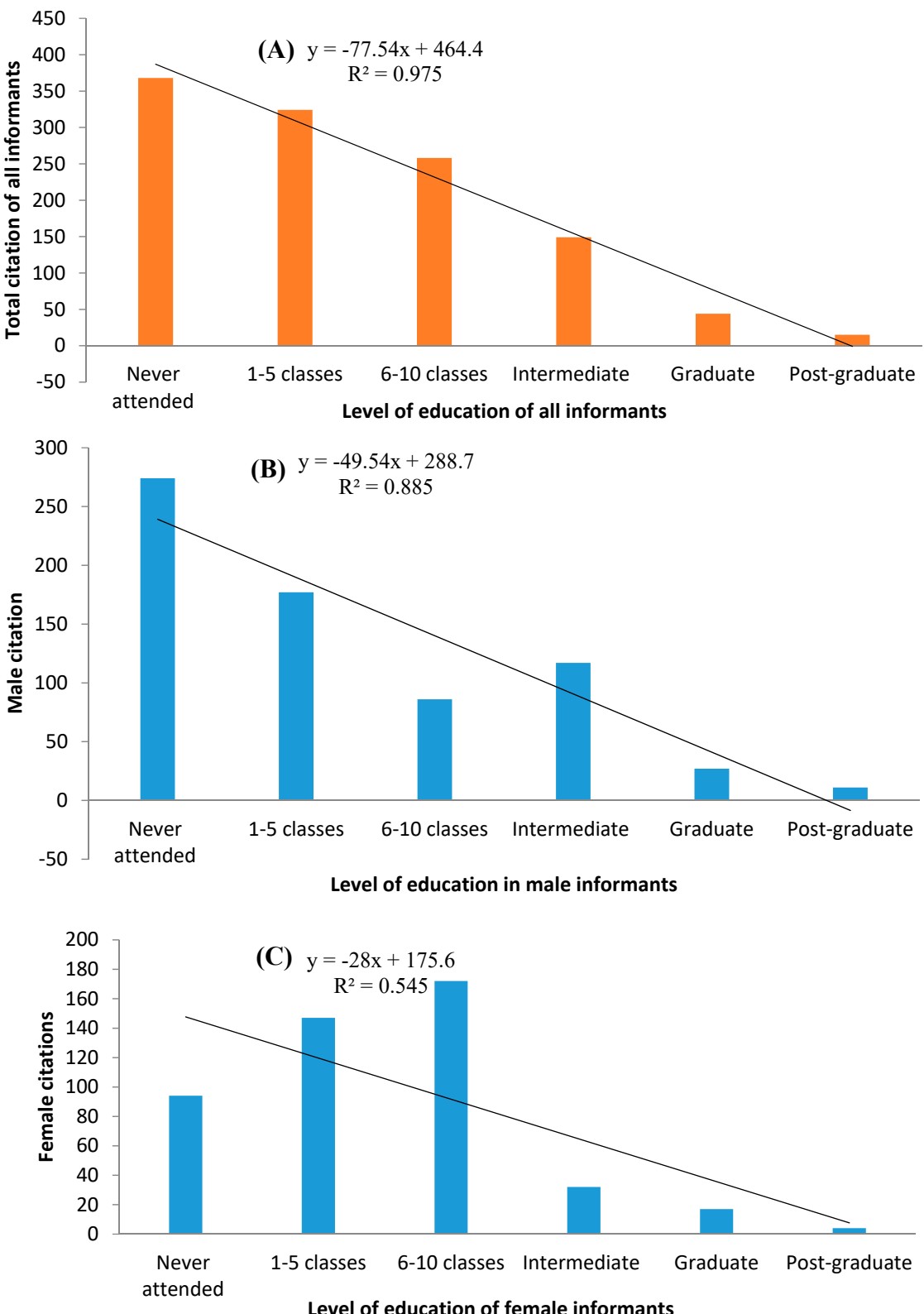

**Figure 3.** (**A–C**) Relative knowledge of informants (citation of medicinal plants) vs. level of education in the study area: (**A**) all users, (**B**) male informants, (**C**) female informants.

### 3.2. Floristic Analysis Of families of Medicinal Plants

From the ethnobiological investigations, a total of 121 plant species belonging to 105 genera and 53 families were recorded to be used as a medicine, which was commonly used by most of the *Duggas, Pahari, Punjabi*, and *Gujjars* traditional healers for the treatment of 15 different types of ailments. Out of 121 species of plants, 96.6% species were categorized as dicotyledons, 2.4% species were monocotyledon, and 0.8% species were non-flowering groups. Most of these taxa were recorded to be growing wild in the ecosystem of the JWS and the adjoining hills and mountains. The most prominent families (Figure 4) were Fabaceae (10 genera and 11 species; 9.09%), Asteraceae (8 genera and 8 species; 6.62%), Euphorbiaceae (5 genera and 6 species), Caesalpinaceae (4 genera and 6 species), Apocynaceae (4 genera and 6 species), Lamiaceae (3 genera and 5 species), Malvaceae (4 genera and 5 species), Moraceae (4 genera and 5 species), Amaranthaceae (4 genera and 4 species), Convolvulaceae (4 genera and 4 species), and Solanaceae (4 genera and 4 species). For each recorded species from the JWS, we provided the scientific (botanical) name, family, voucher specimen number, local (Dogri) name, life-form (tree, shrub, herb, liana), part (s) used (whole plant, bark, root, rhizome, seed, leaves, flower, fruit, latex, resin), ailments treated (disease category), mode of usage (paste, decoction, oral, chewed, powder, juice), use-value, and relative importance (Table 4).

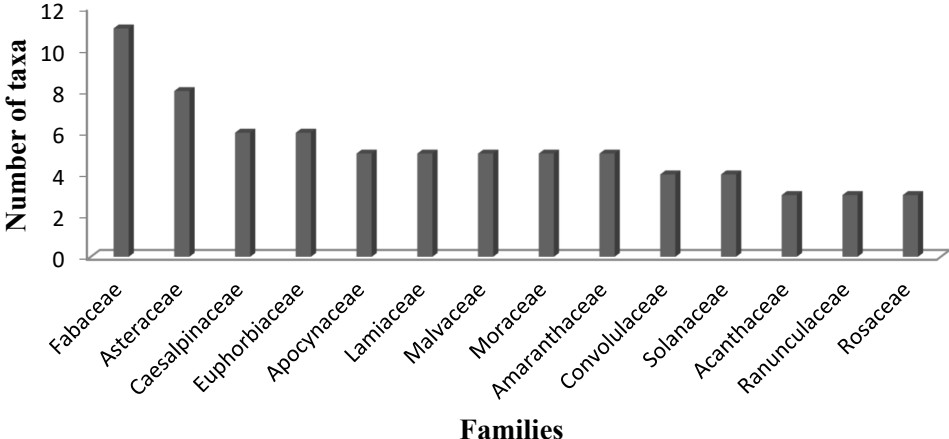

**Figure 4.** Dominant angiosperm families in the study area.

Fabaceae, Asteraceae, Euphorbiaceae, Apocynaceae, Lamiaceae, Rosaceae, Convolvulaceae, and Solanaceae are the most commonly used ethnomedicinal plant families in the Himalayas, and Rao et al. [83] are of the opinion that the reason could be that the plants of these families were well-known among the hilly indigenous communities for their active chemical constituents and easy availability in all climatic conditions. In line with this view, various pharmacopeia also mentioned the importance of these families among the local people, and the reason for this could be that these plant species are a rich source of alkaloids and flavonoids needed for growth and as building blocks of the body [1,57]. Various floras and recent inventories works [87] also mentioned the dominance of Asteraceae, Fabaceae, Lamiaceae, Liliaceae, Solanaceae, Ranunculaceae, Rosaceae, and Polygonaceae, and the reason for this could be the wider distribution range, abundance, and predominant herbaceous habit. The Lamiaceae (mint family) members possess a wide variety of aromatic, medicinal, and ornamental plants producing volatile (essential) oils that are used in the pharmaceutical industry [88]; their chemical constituents, such as menthol, eugenol, geraniol, citral, methyl chavicol, and several other constituents, are intended for the flavor and fragrance industries to produce medicine, perfumes, and cosmetics [89]. This family is known for effective pain modulation with potential analgesic or antinociceptive effects, and mostly includes aromatic medicinal spices such as basil, mint, oregano, and rosemary, and several other species [90,91]. Leguminosae or Fabaceae (or the pea family) coupled with Asteraceae or composite (or sunflower family), with a large number of species distributed in all climates, are more likely to become used by the local

population across the globe [92]. Fabaceae represents the third largest family, and is known to contain various bioactive constituents with potential pharmacological and toxicological effects [93]; most of the species grow in all climates, and various members have been found to possess antimicrobial and antioxidant activities due to the presence of active phytochemicals [94]. Asteraceae represents the largest family of flowering plants, which have been reported to possess several biological functions (antitumor, antibacterial, antifungal, anti-inflammatory) and are known for different classes of chemical compounds, such as polyphenols, flavonoids, and diterpenoids [95,96]. In the study area, members of Moraceae (the fig family) such as *Ficus benghalensis* L. and *F. religiosa* L. were recorded, and these species have been reported to have a wide variety of chemical constituents, with potential biological activities such as being anti-inflammatory [97].

### 3.3. Medicinal Plant Species, Use-Reports (UR), and Associated Knowledge

Among the 121 species mentioned by respondents from the studied villages, analysis of the contribution of the 12 villages to the UR shows that Jasrota village accounts for the higher percentage (Rf = 90.08%), with 943 URs representing 109 species of plants as medicine, followed by Chelak village (Rf = 83.47%, 101 species, 867 URs) and Amala village (Rf = 64.46%, 78 species, 579 URs) based on the proportion of informants interviewed (Table 1). Out of 113 informants, 87.6% reported that these plants were used in self-care. In total, we obtained 4987 URs associated with 121 different species of plants (Table 3).

The use-reports (UR) represent the relative significance of medicinal plants for certain categorized of uses for the treatment of ailments. High values were considered the most important species among the local people. Out of 121 plant species, the 15 most important medicinal plants with the highest use-reports indicating the popularity of the plant species were *Achyranthes aspera* L., *Acorus calamus* L., *Adiantum capillus-veneris* L., *Aloe vera* (L.) Burm.f., *Boerhavia diffusa* L., *Cassia fistula* L., *Eclipta prostrata* L. (L.), *Euphorbia hirta* L., *Justicia dhatoda* L., *Mentha arvensis* L., *Mentha longifolia* (L.) Huds, *Phyllanthus emblica* L., *Tinospora cordifolia* (Willd.) Hook.f. and Thomson, and *Vitex negundo* L. (Table 3). The five most commonly used medicinal plant species with the highest UR (>110)) were *A. aspera* (126UR), *M. longifolia* (119UR), *E. prostrata* (116UR), *E. hirta* (115UR), and *A. vera* (113UR). The highly user-reported species are most frequently harvested for medicinal use and purpose, and these important species need conservation priority. Besides these, other important medicinal plants species among the local herbal healers with multiple usages and high URs were *B. diffusa* (109), *M. arvensis* (106), *T. cordifolia* (105), *J. adhatoda* (104), *A. capillus-veneris* (103), *A. calamus* (99), *C. fistula* (91), *p. emblica* (90), and *V. negundo* (88). Most of these species were frequently used by local *Paharis, Gujjars,* and *Duggers* in Kathua district and other parts of J&K and India for the treatment of various ailments. The plants with a very low UR were *Taxodium distichum* (L.) Rich, *Ranunculus laetus* Wall., *Phoenix sylvestris* (L.) Roxb., and *Albizia lebbeck* (L.) Benth.; all these species were reported with only six URs (out of 113 informants). It has been observed that, although these species were not common in locals, the local healers and ayurvedic doctors often use them in their herbal formulation.

In general, plant species such as *A. aspera*, *A. calamus*, *A. capillus-veneris*, *B. diffusa*, *C. fistula*, *E. prostrata*, *E. hirta*, *J. adhatoda*, *M. arvensis*, *M. longifolia*, *T. cordifolia*, and *V. negundo* were found to be growing abundantly in the Himalayas. Several research studies in J&K (Kathua) [26,29], as well as another mountainous region of India [25,34,41,98], reported high use-values of medicinal plants such as *M. longifolia*, *A. capillus-veneris*, *M. arvensis*, *A. calamus*, *B. diffusa*, *T. cordifolia*, and *A. aspera*. Studies have reported that *M. longifolia* and *M. arvensis* are used as an anti-spasmodic, carminative, anti-septic, anti-peptic ulcer agent, cooling effect medicine, and anti-diarrhea, as well as to cure indigestion and flatulence [41,63].

### 3.4. Life-Form and Plants Parts Used for Medicinal Value

Various growth forms (life-forms) were used as ethnomedicine by local healers (Figure 5). Based on life forms (habit), herbs were the primary source of medicine, and the contribution of

the herbaceous community was the maximum represented by 46.28% of the total 121 investigated species. Other contributors or plant communities were trees (27.27%), shrubs (20.66%), and lianas (5.79%). The plants mentioned in the study were growing in the wild, and the informants (male/female) generally collect them from nearby locations, either from inside the wildlife sanctuary or other forest areas or from wastelands; occasionally, they collect and grow them in a garden or field and judiciously use them depending on their requirements. Some of the forest medicinal plants were collected by the local people for their economic upliftment; they are sold in local (village) weekly markets, and the herbal users or ayurvedic doctors purchase from them in the market and store them for future utilization.

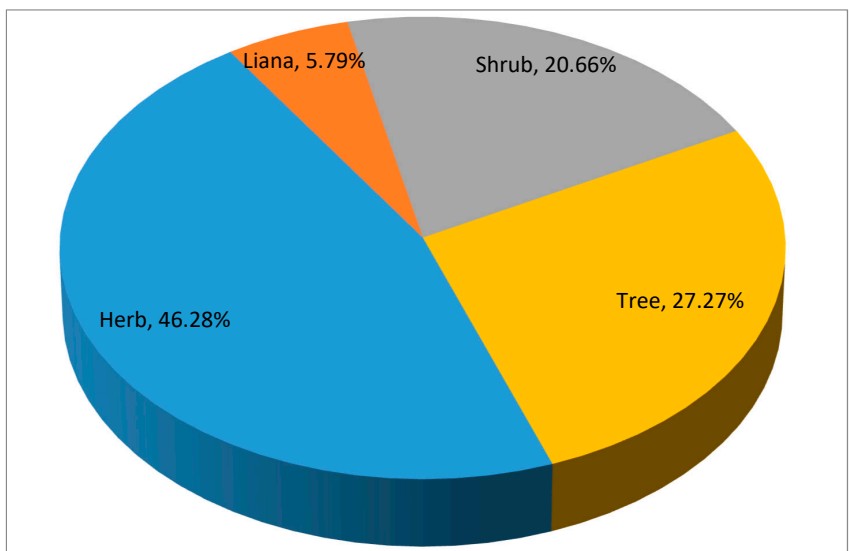

**Figure 5.** Life forms of the recorded medicinal plants in the study area.

The frequent use of herbs as medicine among the local people was the result of the abundant availability of herbaceous plants in nearby locations, either in the forested zone (protected or unprotected) or in wastelands. This result agrees with several other plant inventories studies in India and other adjoining countries, which indicated the dominance of herbaceous species in the natural ecosystems [98–104]. J&K and particularly the district Kathua harbor the maximum number of herbs as compared to various other growth forms, such as trees, shrubs, and lianas [105–111]. Herbs are often reported to have a high content of secondary metabolites and bioactive compounds [107–109]; therefore, their medicinal action in human health care was found to be more effective than that of shrubs and trees [110]. They can also grow more commonly along roadsides and in home-gardens and are easily accessible [111]. The dominance of herbs in ethnomedicinal floral diversity could be due to them being more easily available in the nearby areas, or it may be due to the short life-spans, as local people look for alternate herbs of similar effect as compared to trees and shrubs [112].

In the study area, various parts (bark, root, rhizome, seed, leaves, flower, fruit, latex, resin) including the whole plant of the investigated species were used by the local people as medicine in the treatment of commonly occurring diseases. Among the different plant parts, leaves (43.80%) were the most frequently used part among the total documented species from the study area. This plant part was then followed by whole plants (15.70%), roots (10.74%), fruits (9.09%), flowers (5.79%), seeds (3.91%), bark of stems (4.13%), latexes (2.48%), rhizomes (1.65%), resins (1.65%) and heart-wood (0.78%) (Figure 6). Based on the percentage of use-reports for the respective ailment category mentioned by the informant, the most frequently treated gastrointestinal ailments (GIA) indicate 24.55% of the whole plants (WP), 33.08% of the underground parts (UGP), 47.18% of the barks (BR), 24.39% of the aerial parts (AEP), and 1.37% of the latex/resins (LR) (Table 3).

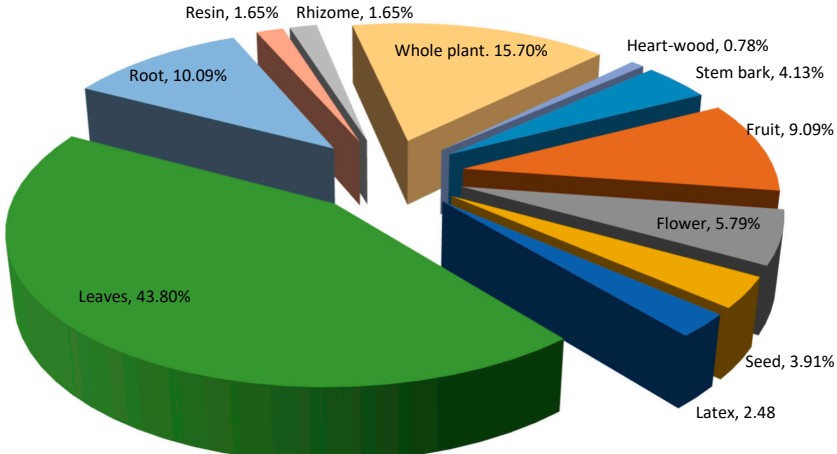

**Figure 6.** Plant parts (percentage) used for the preparation of medicine in JWS.

Several other similar studies in Indian states and other countries [35,84,103,113–118] have also supported that leaves were the most dominant plant part used for the preparation of herbal drugs among many indigenous communities, and the reason for this could be that leaves are collected more easily than underground parts (rhizomes, tubers, roots) and flowers or fruits [119,120]. However, from the scientific point of view leaves are the most active parts of a plant involved in photosynthesis and are involved mostly in the production of secondary metabolites [121]. These findings were also supported by Castellani [122], who believed that soft plant parts (leaves, buds, flowers) contain several rich sources of bioactive volatile components and active principles. Similarly, few research studies in J&K and, in particular, Kathua [123,124] have also reported leaves as the most used plant part used because of their potency and faster regeneration than other parts. The whole plants or roots were also reported frequently used in the preparation of herbal medicine [25,34]. Giday et al. [111] and Jan et al. [72] reported that the consumption of leaves as compared to other plant parts for therapeutic value is more sustainable and helps in the conservation of species.

### 3.5. Herbal Remedies and Ingredients Added

The herbal formulations and administration mode used by herbalist practitioners in the study area varies and depends on the number of plants or their parts used and the ingredients added as additives. In some remedies, single taxa or plant parts are used; for example, for the treatment of ringworm, the decoction obtained from the leaves of *Argemone mexicana* commonly locally called "peelikandiari" is used. Similarly, for curing diarrhea *Bombax ceiba* root is boiled and the decoction obtained is drunk to get relief from diarrhea. On the other hand, in most of the cases two or more plant taxa or their parts (leaves, roots, fruits, flowers, seeds, stems, bark, rhizomes, tubers, resin, heart-wood, or whole plant) are used by the *Duggar, Pahari, Punjabi*, and *Gujjar* communities in the preparation of herbal remedies for curing same ailments or for multiple disorders associated with the same body system. The herbal doctors or aged medicinal practicing people of the community believe that one species of plant may perhaps be useful for the treatment or curing of several human problems, which is due to the fact that different classes of bioactive secondary metabolites or compounds responsible for different disorders may be present in different species of the same nature or vice-versa. The findings mentioned in this study were further in line with the studies reported in several previously published accounts [82,86]. A few studies carried out in different parts of India [125,126] have also mentioned that the manifold uses of plants may be attributed to various synergistic reactions where one plant could produce a greater effect than other, which could be the interaction of two or more than two active compounds, or either would have been produced by the same species.

A few studies of Thakur et al. [7] and Teklehaymanot et al. [107] were of the view that plant recipes (decoction, juice, powder, paste, etc.) using one plant or their parts were more common among

local tribes in hilly and mountainous regions. Singh et al. [36] and Ayyanar and Ignacimuthu [86] reported that the poly-herbal recipes, where many plants and their parts were used for preparation, have more curing power (healing power), which is due to the presence of varieties of active compounds for biological (pharmacological) functions. Relevant studies [1,6,7,85] have reported that the suggested dosage by herbal doctors prepared for a particular ailment always differs and depends on the severity of the disorders of body functions, the stage of the disease, and the age of the suffering person; mostly, the liquid extracts, preparations, or decoctions were recommended as a full, 1/2, or 1/4 of a cup or glass, whereas the powder remedies recommended dosages were in terms of one or a 1/2 of a spoon or pinch.

### 3.6. Preparation and Mode of Administration

The preparation of herbal remedies and the utilization of plant parts were grouped into five categories (powder, juice, paste, decoction, chew) (Figure 7). Of these, decoction (33.88%) was the most commonly used method of drug/medicine preparation. It was then followed by using as juice (22.31%), paste (21.49%), powder (18.18%), and chewed/raw (4.13%). The decoctions were considered as squeezed materials that were obtained by boiling the parts of plants in water until the water volume was reduced to half or one-third depending on parts used and requirement, and consumed as drunk to heal internal injuries. The most common method of preparation was extracted in hot or cold water, and the mode of administration was oral rather than topical or other forms. Decoction as the primary method of preparation method was supported by the recent studies of Dapar et al. [88], where Manobo tribe of Agusan del Sur (Philippines) uses the decoction method as most common in curing diseases, followed other modes such as powder, poultice, juice extract, paste, or infusion. This finding indicates decoction as the highest frequency for the preparation and administration of herbal drugs, and is supported by previous several ethnobotanical investigations across the globe. This result is also contrary to the previous reports on ethnic tribes such as the Ati Negrito community of the Visayas, Philippines [127]. The powder as an herbal cure was prepared from the dried plant parts, which may be roots, barks, tubers, rhizomes, and stems, and the method employed was grinding to a fine powder in the study area; this result was also supported by Ayyanar and Ignacimuthu [86].

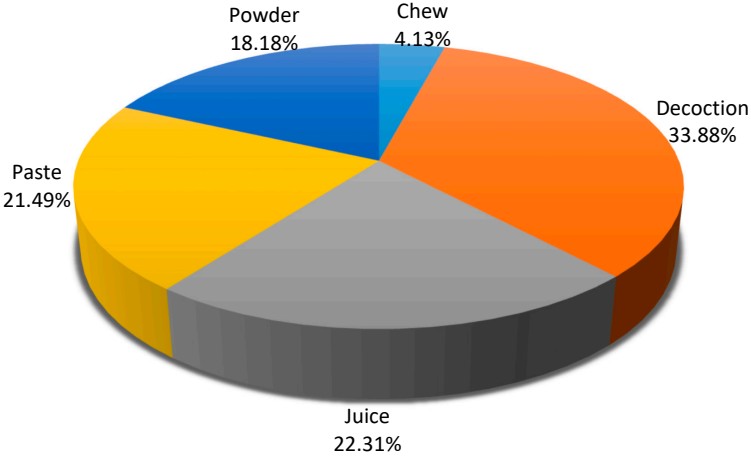

**Figure 7.** Categories of local tribes' mode of utilization (percentage) for the preparation of herbal medicine.

The paste mode of administration is yet another very common practice for the treatment of ailments in the study area, which is prepared by grinding the fresh or dried plant parts (leaves, stem, flowers, bark, fruit) with water, oil, honey, or lime. This study was also supported by previous researchers on other tribal communities in India [85,128] and other parts of the world [129]. Abe and Ohtani [130] recorded ethnobotanical studies on the Ivatan community in Luzon and mentioned paste as the most common way of treating external ailments, which mostly dealt with diseases such as skin infections, wounds (cuts, boils, burns), rheumatism, poison bites (snake, scorpion, insects), heel cracks, and headaches

in case of malaria fever. The external administration was considered as safe as compared to internal ailment treatments. In cases of external diseases and illnesses, more prolonged mustard, coconut, or castor oil infusions with paste or powder were often applied. Occasionally, fine powders of medicinal plants were applied to wounds or cuts to stop bleeding. Chewing of some plant parts is less predominant in the study area; however, some medicines were given orally, and this study is in agreement with some others conducted elsewhere in the world [131,132]. It has been observed that the herbal healers, before giving treatment to any patient, deeply observe the health conditions, age, and physical status of the patients, and then they prescribe or gave a dosage of an already-prepared medicine.

### 3.7. Informants Consensus factor (ICF) and Ailment Category

Generally, the ICF indicates agreement among informants on the utilization of plant taxa for a particular purpose or disease category, and various ailments categories depended on the availability of the plant species in the investigated area. To use ICF, we classified the ailments into 15 broad categories, and the results obtained from the study area on various ailments categories are mentioned in Table 3. The ICF values in the study areas ranged from 0.667 to 0.974. The dental care category (ICF = 0.974) had the maximum and the cancerous (ICF = 0.667) had the minimum consensus between the informants. As mentioned in Table 3, the ICFs for the use categories are relatively high, and this suggested the exchange of information among the local people of the study area on medicinal plant species, which indicates the apparent efficacy of reported species.

The plants used to treat the gastrointestinal ailments (GIA) were the most frequently reported (1314 URs, 26.65%, 55 species), followed by the dermatological infections (DID) (946 URs, 18.97%, 38 species) (Table 3). Other ailment categories with more than 200 use-reports (URs) were respiratory system diseases (RSD) (592 URs, 24 species), skeleton-muscular system disorders (SMSDs) (482 URs, 21 species), genitor-urinary ailments (GUAs) (352 URs, 19 species), and general health care (GHC) (246 URs, 14 species) (Figure 8).These results showed that the exchange of information could be evident among the different communities on their medicinal plant uses and practices. Adding more, the different ailments that arise may be due to the nature of poor economic activities, health, and sewage facilities.

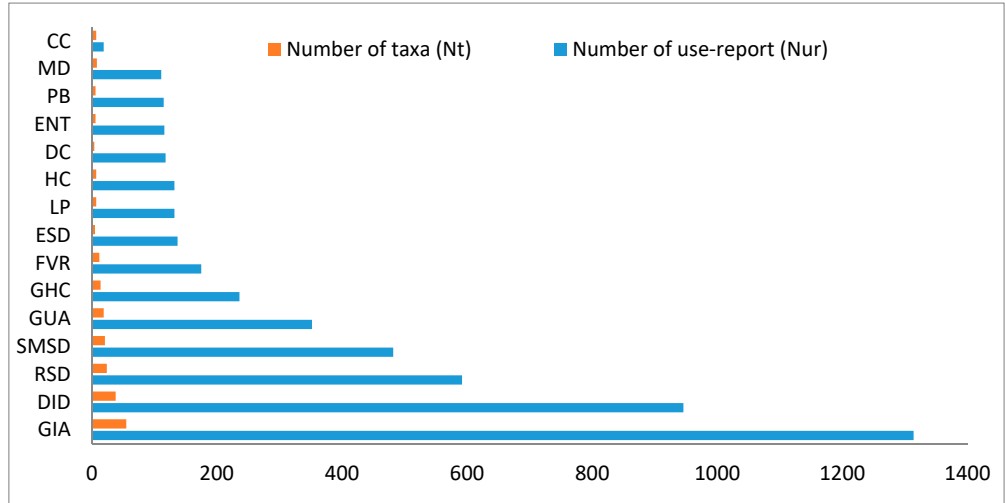

**Figure 8.** Categories of ailments treated by the local Duggars, Paharis, Punjabis, and Gujjars tribes arranged by the number of taxa and number of use-reports (CC—cancerous; DC—dental care; DID—dermatological infections (or diseases); ENT—ear, nose, and eye problems; ESD—endocrinal system disorders; FVR—fevers; GIA—gastro-intestinal ailments; GUA—gento-urinary ailments; HC—hair care; LP—liver problem; RSD—respiratory system diseases; PB—poisonous bites; SMSD—skeleto-muscular system disorders; GHC—general health care; MD—mouth disorders).

Among the 15 use categories, 5 categories—namely, diseases of the gastro-intestines, diseases of the skin (dermatological), respiratory diseases, endocrine system disorders, and problems with dental care—had the highest value of more than 0.959, and the most plants employed for the treatment were 55, 38, 24, 5, and 4 species, respectively. This study was found to be in agreement with the previously published studies of Andra-Cetto [132], Ayyar and Ignacimuthu [86], Weckerle et al. [46], and Dapar et al. [88]. Skeleto-muscular system disorders had an ICF of 0.958, but this ailment category ranks fourth in the number of use-reports (482) and the number of species (21) attributed to this category, and this may be due to the lack of proper communication among the informants, as different communities reside the area; the study was found to be in line with the earlier studies undertaken by Ragupathy et al. [133], Rokaya et al. [134], Rao et al. [41], and Singh et al. [25,85]. Illustrating this further, Heinrich et al. [54] reported that gastro-intestinal problems, skin infections (or disorders), and respiratory system diseases had a high ICF in a study conducted on the Maya, Nahua, and Zapotec people in Mexico. The study is also in line with the research of Weckerle et al. [46], who studied the valley of Juruena (Legal Amazon, Brazil) and reported that gastro-intestinal disorders, skin infections, and respiratory problems can be treated easily by utilizing locally available plant resources. In J&K, Bhatia et al. [84] reported that gastro-intestinal diseases and dermatological problems were the main disease categories in Udhampur district, and the reason responsible could be the poor sanitation facilities. Rao et al. [41] also reported that the poor sanitation facilities in hilly areas and mountainous regions are responsible for more gastro-intestinal diseases. Diabetes (an endocrine disorder) and jaundice (liver problem) are yet other importance ailments found to be common in the study area. Other similar studies were undertaken by Pandikumar et al. [135] on the Paliyar tribe in the Theni district, Ayyanar and Ignacimuthu [86] on the Kani tribe in the Tirunelveli Hills of Western Ghat, Ragupathy et al. [133] on the Malasar tribe in the Coimbatore district, and Ragupathy and Newmaster [136] on the Irulas tribe in the Thanjavur district (Tamil Nadu). Rao et al. [41] on the ethnobotany of the Kathua district also supported diabetes and jaundice as the most prevalent disease in hills and mountains and stated that several local plants and formulations were available to local people to help in the curing of these diseases. The increasing prevalence of diabetes may be due to changes in diet, reductions in physical activity, and overweight and obesity [137].

The ICF values in the study area ranged between 0.667 and 0.974. These values are in agreement with those of earlier studies conducted elsewhere in India [41,103,136] and other countries [109,138]. The high ICF indicates the high level of consensus among the informants for a particular disease category and was supported by Sharma et al. [139] and Dapar et al. [88]. The high level of consensus about the utilization of medicinal plants for the treatment prevalent in the study area suggests that the traditional knowledge and ethnomedicinal usages of plant species are currently in practice, and this is supported by several studies [140].

## 4. Conclusions

This study highlights the need for the documentation of targeted unexplored areas of biocultural diversity rich in ethnobotanical knowledge and the ethnopharmacological practices of the local communities. The Himalayan *Duggar, Pahari, Punjabi*, and *Gujjar* people residing in and around the JWS protected area were focused on using the available natural plant resources. In conclusion, the results revealed a high diversity of medicinal species used by the people, with 121 taxa utilized in 15 ailment categories. Like other ethnolinguistic indigenous communities, the traditional knowledge of this belt may get lost or forgotten due to acculturation, migration for high-class education and jobs, and the lesser interest of the younger population in response to modernization and their education level. The species of plants with a high fidelity level, use-value, and relative importance indicated the deposition of several valuable phytochemical compounds in different parts of the plant, and future natural product isolation from this studied species will help in the new discovery of bioactive chemicals for drugs and new herbal formulations from plants to cure various diseases in the future. The plants with 100% fidelity levels need to be targeted for future analysis for associated pharmacological

studies. Although the local medical herbalists claimed a high level of efficacy, there is a need for the scientific standardization of their recipe preparation techniques, the dosage, administration methods, and the accuracy of disease diagnosis. This study asserted the scientific validation of all documented ethnomedicinal plant species for their safety and efficacy against the ailments through more chemical and pharmacological research. Ecological studies and the sustainable management of biodiversity present in the protected areas through conservational efforts needed to protect the depleting plant resources. The findings mentioned in this study will serve as future reference material for research in the field of systematic, biochemical, and pharmacological studies. The findings of this study are promising regarding new potential therapeutic agents for human healthcare.

**Author Contributions:** Conceptualization, B.S. (Bishander Singh) proposed the research study for his Ph.D. degree, carried out the fieldwork; writing—original draft preparation, A.K., B.S. (Bishander Singh), S.S., O.S., and M.N.B. assisted with the species identification and authentication; writing—review and editing, B.S. (Bishander Singh) and B.S. (Bikarma Singh) evaluated the data of the fieldwork for inclusion in the manuscript; supervision, B.S. (Bikarma Singh), A.K. and C.M.M. reviewed, analyzed, and gave critical comments. All authors have read and agreed to the published version of the manuscript.

**Funding:** This research received no external funding, but it's a part Ph.D. work of the first author.

**Acknowledgments:** The authors are thankful to the Head, Department of Botany, Veer Kunwar Singh University, Ara, Bihar, and Director of CSIR-Indian Institute of Integrative Medicine, Jammu, for providing the herbarium facilities and moral support, and are also thankful to the local people for revealing their traditional knowledge.

**Conflicts of Interest:** The authors clearly declare that they have no conflict of interest.

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
