# Peer review of "Exploring Plant-Based Ethnomedicine and Quantitative Ethnopharmacology: Medicinal Plants Utilized by the Population of Jasrota Hill in Western Himalaya"

_sustainability, doi:10.3390/su12187526_

Round 1

Reviewer 1 Report

This manuscript focuses on the plant-based medicine of the communities living in the Jasrota hills, Northern India. The authors documented this knowledge with the aim to discover new potential drugs.

The manuscript is generally clearly written and informative, yet few major points can be improved: 

  • the current categorization of the ailments can be improved (e.g. n my opinion, halitosis should not be in the respiratory diseases for instance, while it could fit in the disorders of the digestive system). For an internationally recognized categorization, I strongly advise you to follow https://icd.who.int/browse11/l-m/en.
  • The authors mentioned different minorities living in the area. Can the belonging to one or another minority affect the ethnomedicinal knowledge? What is the share of minorities in your sample (n=113)?

  • Who are the resource persons? How have you selected them? How have you selected the participants?

Minor points:

Line 45: "The ethnobotanical plant has a very base in Ayurveda" what does this mean?

Lines 64-67: bordering is a bit redundant here in the intro. I'd suggest deleting these lines.

Line 139: could you please include an explanation of "shivalik"

Line 226: I would suggest relying on internationally recognized codes of ethics (e.g. https://www.ethnobiology.net/what-we-do/core-programs/ise-ethics-program/code-of-ethics/code-in-english/#:~:text=The%20Code%20of%20Ethics%20of,ethnobiological%20research%20and%20related%20activities.)

Line 269: age correlation is mentioned here in the methods, but I cannot find this point in the results section. 

Line 706: climate change comes a bit as a surprise here as it was never mentioned in the text

Line 707: how can your study contribute to reducing the loss of indigenous knowledge?

Fig 1: make J&K explicit

Author Response

Reviewer # 1:

This manuscript focuses on the plant-based medicine of the communities living in the Jasrota hills, Northern India. The authors documented this knowledge with the aim to discover new potential drugs. The manuscript is generally clearly written and informative, yet few major points can be improved:

Comments Response
1. The current categorization of the ailments can be improved (e.g. n my opinion, halitosis should not be in the respiratory diseases for instance, while it could fit in the disorders of the digestive system). For an internationally recognized categorization, I strongly advise you to follow https://icd.who.int/browse11/l-m/en Reply:  As per reviewer comments and suggestions, the word ‘halitosis’ is shifted from the respiratory category to the Gastro-intestinal ailments (GIA) category which contains disorder of the digestive system, and subsequently, changes were made throughout in the revised manuscript.
2. The authors mentioned different minorities living in the area. Can the belonging to one or another minority affect the ethnomedicinal knowledge? What is the share of minorities in your sample (n=113)? Reply: The minorities living in the study area were mostly the migrant labourers and data were not collected from them. The share of minorities n=113 is nil/zero as they were not involved in this study.
3. Who are the resource persons? How have you selected them? How have you selected the participants? Reply: As per the reviewer comment, the resource persons for this particular study were the local hakims, medicine men, ayurvedic medicinal man, elders, and sarpanch/head of village person who has knowledge of plants and their use. Information collected from all participants was selected by group discussion, interviews, etc
4. Line 45: "The ethnobotanical plant has a very base in Ayurveda" what does this mean? Reply: Himalaya region is a home of several nomadic indigenous tribes, and Ayurvedic medicine is the indigenous Indian system of medical practice where locally available plants are used for the treatment of commonly seasonal occurring diseases. Line no. 45 is rephrased for better understanding and meaning
5.Lines 64-67: bordering is a bit redundant here in the intro. I'd suggest deleting these lines. Reply: As per reviewer suggestions, lines 64-69 were deleted in the revised manuscript.

6.Line 139: could you please include an explanation of "shivalik"

Reply: Line 139 is reframed and a brief explanation of ‘Shivalik’ is provided in the revised manuscript as suggested by the reviewer.
7.Line 226: I would suggest relying on internationally recognized codes of ethics (e.g. https://www.ethnobiology.net/what-we-do/core-programs/ise-ethics-program/code-of-ethics/code-in-upenglish/#:~:text=The%20Code%20of%20Ethics%20of, ethnobiological%20research%20and%20related%20activities Reply: As per nice suggestions and links provided, line no 226 is reframed and the reference of the International Society of Ethnobiology (https://www.ethnobiology.net) incorporated in the revised manuscript
8.Line 269: age correlation is mentioned here in the methods, but I cannot find this point in the results section Reply: Age correlation data is mentioned in the result (subhead category 3.1. Characteristics of informants) as ‘The spearman’s rank correlation analysis between age and number of citations of medicinal plants (r=0.257; p<0.001) were significant for the investigated people” and explained

9.Line 706: climate change comes a bit as a surprise here as it was never mentioned in the text

10.Line 707: how can your study contribute to reducing the loss of indigenous knowledge?

  1. Fig 1: make J&K explicit

Reply: As per suggestion, the word ‘climate change’ is deleted from the manuscript in the revised version.

Reply: The line is deleted from the manuscript in the revised version

Reply: ‘J&K’ is explicit as Jammu and Kashmir in the revised version

Reviewer 2 Report

The article titled “Exploring plant-based ethnomedicine and quantitative ethnopharmacology…” attempted to highlight the prevailing local knowledge on medicinal properties of plant species. The authors conducted a systematic survey by including various age groups of local healers. The quantitative parameters such as use-reports (UR) and informant consensus factor (ICF) have been included in the study. Overall, the manuscript seems appealing because of a) broader participants pool and b) a decent number of interviews/surveys are conducted. Nonetheless, the manuscript needs amendments and technical editing to address issues in the article. I strongly suggest statistical analyses to strengthen the study. 

  1. The authors have listed several electronic resources for their review of the literature. However, the authors have omitted an important resource on medicinal plants which include botanical description and Photogallery of herbariums is the “Encyclopedia on Indian Medicinal Plants” by ENVIS center on Medicinal plants and FRLHT (http://envis.frlht.org/). I strongly suggest the authors identify and report any novel or new therapeutic uses of plant species discovered in the study. The pieces of literature reviews/electronic resources can be used for such purposes.
  2. The plant species are broadly categorized to botanical families. However, there is no report on endemic plants to the surveyed areas. It would be interesting to know if endemic plants are explored for special therapeutics use.
  3. The authors cover a wide geographical area and a diverse botanical composition in their survey. I suggest the authors use statistical analyses such multi-dimensional scaling (MDS) to a) decompose the variation in plant-use and participants into distances or dissimilarities b) differences in environmental factors like elevation, local climate, seasons and livelihood to assess if these characteristics explains the plant-use spaces. 
  4. Minor comments:

Line 119: correct ‘ad’ to ‘and’

Line 200: please rephrase the sentence

Line 553: please rephrase the sentence.

Author Response

The article titled “Exploring plant-based ethnomedicine and quantitative ethnopharmacology…” attempted to highlight the prevailing local knowledge on medicinal properties of plant species.

Comments Response

The authors conducted a systematic survey by including various age groups of local healers. The quantitative parameters such as use-reports (UR) and informant consensus factor (ICF) have been included in the study. Overall, the manuscript seems appealing because of a) broader participants pool and b) a decent number of interviews/surveys are conducted. Nonetheless, the manuscript needs amendments and technical editing to address issues in the article. I strongly suggest statistical analyses to strengthen the study. 

The authors have listed several electronic resources for their review of the literature. However, the authors have omitted an important resource on medicinal plants which include botanical description and Photogallery of herbariums is the “Encyclopedia on Indian Medicinal Plants” by ENVIS center on Medicinal plants and FRLHT (http://envis.frlht.org/). I strongly suggest the authors identify and report any novel or new therapeutic uses of plant species discovered in the study. The pieces of literature reviews/electronic resources can be used for such purposes.  The plant species are broadly categorized to botanical families. However, there is no report on endemic plants to the surveyed areas. It would be interesting to know if endemic plants are explored for special therapeutics use. The authors cover a wide geographical area and a diverse botanical composition in their survey. I suggest the authors use statistical analyses such multi-dimensional scaling (MDS) to a) decompose the variation in plant-use and participants into distances or dissimilarities b) differences in environmental factors like elevation, local climate, seasons and livelihood to assess if these characteristics explains the plant-use spaces. 

Reply:  The authors would like to thanks anonymous reviewers for his recommendable critical comments and beautiful suggestion on our manuscript, and we accepted all corrections in the revised manuscript. We have incorporated some statistical analysis and some we reduce considering the length of the manuscript. With comparison to the previous submission, the length of the manuscript reduced to half i.e. from 115 pages to 77.

As per suggestion and comment, we have included “Encyclopedia on Indian Medicinal Plants” by ENVIS center on Medicinal plants and FRLHT (http://envis.frlht.org/) in the revised version.

As per comment, novel or new therapeutic uses of plant species discovered while studying mentioned in the manuscript.

As per comments, we have not come across any endemic plants of ethnobotanical importance from the study area and hence is not mentioned. Further, we have done some statistical analyses but considering the length of the manuscript, we have not incorporated MDS and other parameters with more focus on plant use in different diseases.
Line 119: correct ‘ad’ to ‘and’ Reply:  Correction of ‘ad’ to ‘and’ was incorporated in the revised manuscript.
Line 200: please rephrase the sentence Reply:  As per the reviewer's comment, the sentence has been rephrased and incorporated in the revised manuscript
Line 553: please rephrase the sentence Reply:  As per the reviewer's comment, the sentence was rephrased and incorporated in the revised manuscript

Reviewer 3 Report

The article “Exploring plant-based ethnomedicine and quantitative ethnopharmacology: Medicinal plants utilized by population of Jasrota Hill in Western Himalaya” was submitted for review. The abstract is composed logically, contains information about the relevance, design, localization of the study, statistical methods of material processing, and description of the study. The study collected information on 121 plant species belonging to 105 genera and 53 families for the treatment of 93 types of diseases. The article is extremely relevant due to its high medical and social importance. Traditional medicine has a deep knowledge of the effectiveness of herbal preparations based on natural ingredients. At the same time, documenting traditional knowledge plays an important role in preserving the thousands of years of experience of peoples. Wild plants of India, as natural resources used by tribal communities in several regions of India, are the subject of numerous ethnobotanical studies. The Himalayas have a large vertical drop and a unique climate, resulting in a wide variety of flora and fauna. Locals depend on plants for common diseases due to the lack of modern medical facilities. Therefore, many researchers want to work on ethnobiology in order to gain the knowledge and culture of indigenous peoples, which local people own for the discovery of new medicines and herbal formulations. In this regard, the present study is devoted to collecting and documenting information on the local use of medicinal plants for the treatment of the most common diseases by local residents. In the section, the material and methods are qualitatively described - the study area and the local population, Data collection, Ethnobotanical data collection, Sampling The section on materials and research methods contains information on design and methods, Informants and interviews with local people, Ailment categories, Data analysis, Informant consensus factor (ICF), Literature reviews, Species identity and library consultation, Electronic databases. The sections results, discussion, conclusion contain all the necessary information, with numerous references to modern literature sources. This study confirmed the validity of all documented ethnomedical plant species for their safety and efficacy against disease. The results mentioned in the study will serve as future reference material for research in the field of systematic, biochemical and pharmacological research, about new potential therapeutic agents for public health.

Conclusion. Based on the high relevance and importance of the research, the qualitative presentation of the material.

Author Response

Reply: We like to thanks anonymous reviewer 3 for very nice appreciations of our manuscript and we are sure that this manuscript is going to serve an important document in the field of conservation of knowledge and application of plants in drug discovery.

Reviewer 4 Report

Dear Authors,

In my opinion the reviewed article is of high value and it requires only small corrections, as given below:

  1. In the chapter: 3. Results and Discussion, 3.1. Characteristics of informants (page 13): it is written that in the study 72 males and 41 females were informants, why such a difference? was it possible to have more equal number of both sexes participating in the study ?
  2. Literature (page 33, position 55, line 881): in my opinion there is error in the year of publication, it should be 2015 perhaps ?
  3. Fig. 6, (page 47) instead of the word: "leave" the term "leaves" should be written
  4. In my opinion in Table 3 there are many symbols, such as Nur %Nur and they should be explained again under the table.

Author Response

Reviewer 4

In my opinion the reviewed article is of high value and it requires only small corrections, as given below:

Comments Response
1. In the chapter: 3. Results and Discussion, 3.1. Characteristics of informants (page 13): it is written that in the study 72 males and 41 females were informants, why such a difference? Was it possible to have more equal number of both sexes participating in the study? Reply: We thanks reviewer comments and suggestions on equality of participants, but also would like to bring into focus that the most women are not allowed to participate in any such occasion as most of them are housewife and allowed to concentrate in the household works. Men are involved to participate all outside works, therefore, the number of male informants is more than female here in this study
2. Literature (page 33, position 55, line 881): in my opinion there is error in the year of publication, it should be 2015 perhaps? Reply: The year of publication ‘2015’ is corrected in the revised manuscript.
3. Fig. 6, (page 47) instead of the word: "leave" the term "leaves" should be written Reply: In Fig.6, the word ‘Leave’ is corrected to ‘Leaves’ in the revised manuscript
4. In my opinion in Table 3 there are many symbols, such as Nur %Nur and they should be explained again under the table Reply: We thank the reviewer for this important nice correction and we have included all abbreviated letters in the revised manuscript

Round 2

Reviewer 1 Report

Accepted in present form

Reviewer 2 Report

No major comments on this version of the manuscript. All  previous concerns/comments have been addressed. 

This manuscript is a resubmission of an earlier submission. The following is a list of the peer review reports and author responses from that submission.